# Down-regulation of a cytokine secreted from peripheral fat bodies improves visual attention while reducing sleep in *Drosophila*

**Deniz Ertekin** , **Leonie Kirszenblat, Richard Faville** , **Bruno van Swinderen** *

Queensland Brain Institute, The University of Queensland, Brisbane, Queensland, Australia

* b.vanswinderen@uq.edu.au

**Data Availability Statement:** All relevant data are within the paper and in its Supporting Information files.

## Abstract

Sleep is vital for survival. Yet under environmentally challenging conditions, such as starvation, animals suppress their need for sleep. Interestingly, starvation-induced sleep loss does not evoke a subsequent sleep rebound. Little is known about how starvation-induced sleep deprivation differs from other types of sleep loss, or why some sleep functions become dispensable during starvation. Here, we demonstrate that down-regulation of the secreted cytokine unpaired 2 (*upd2*) in *Drosophila* flies may mimic a starved-like state. We used a genetic knockdown strategy to investigate the consequences of *upd2* on visual attention and sleep in otherwise well-fed flies, thereby sidestepping the negative side effects of undernourishment. We find that knockdown of *upd2* in the fat body (FB) is sufficient to suppress sleep and promote feeding-related behaviors while also improving selective visual attention. Furthermore, we show that this peripheral signal is integrated in the fly brain via insulin-expressing cells. Together, these findings identify a role for peripheral tissue-to-brain interactions in the simultaneous regulation of sleep quality and attention, to potentially promote adaptive behaviors necessary for survival in hungry animals.

## Introduction

Behavioral decisions in animals are formed by integrating internal states with external stimuli and prior experience. The need for sleep and food are two such internal states and satisfying both of these homeostatic processes seems equally important for survival [1–5]. Yet sleeping and feeding are also mutually exclusive: they cannot happen at the same time. Under environmentally challenging conditions, mutually exclusive behaviors therefore need to be prioritized in order to maximize survival.

Both sleep and feeding regulation have been extensively studied in different animal models, as well as in humans [6–12]. Yet how their pathways intersect and influence each other remains unclear. Given the alarming increase in the number of people with both sleep and metabolic disorders [13–15], to understand how these two processes interact at the level of neural circuits and molecular pathways is of significant interest. *Drosophila melanogaster* has been a pivotal model system to study both sleep and feeding regulation [9,16–23]. Sleep in flies

**Funding:** This work was supported by two grants from the National Health and Medical Research Council of Australia (https://www.nhmrc.gov.au/), GNT1065713 and GNT1164499, to BVS. The funders had no role in study design, data collection and analysis, decision to publish, or preparation of the manuscript.

**Competing interests:** The authors have declared that no competing interests exist.

**Abbreviations:** ARC, Activity Recording Capillary; BDSC, Bloomington Drosophila Stock Center; Café, capillary feeding; CaLexA, calcium-dependent nuclear import of LexA; DART, *Drosophila* arousal tracking; dome, *domeless*; EB, ellipsoid body; FB, fat body; Ilp2, insulin-like peptide 2; IPC, insulin-producing cell; LED, light-emitting diode; LHLK, lateral horn leucokinin; Npf, neuropeptide F; NREM, non-rapid eye movement; PI, pars intercerebralis; RNAi, RNA interference; upd2, unpaired 2.

has been shown to fulfill key criteria for identifying sleep in other animals, such as increased arousal thresholds and homeostatic regulation [24,25], so the fly is a promising avenue for understanding sleep and feeding regulation at a circuit level. Additionally, cognitive readouts such as visual attention paradigms are increasingly available for *Drosophila* research [26–28], providing relevant functional assessments of manipulations that could impact sleep and feeding.

Generally, the effect of feeding on sleep has been studied by altering dietary components, or by more severe interventions such as starvation [29–32]. However, studying this relationship via nutritional manipulations introduces numerous secondary factors (e.g., metabolic processes or energy levels), confounding any analysis of potential interactions between satiety/starvation signals and sleep processes. We therefore decided to utilize a genetic strategy in *Drosophila* to down-regulate a cytokine secreted from the fly fat body (FB), unpaired 2 (*upd2*) [33,34]. By down-regulating *upd2* and its receptor *domeless* (*dome*) [35,36], we aimed to potentially mimic a "starved" state in flies, which allowed us to assess the effect of this metabolic signaling pathway on sleep and attention simultaneously. *upd2* has been suggested as a candidate ortholog for vertebrate leptin, and has similar structure to type-I cytokines [37,38]. Similar to leptin, secretion of *upd2* is dependent on nutrient intake [37,39], and it is secreted from the fly counterpart of adipocytes, or FBs.

Starvation has several consequences on the behavior of animals, with one of the most striking ones being suppression of sleep [29,32,40–43]. Normally, sleep deprivation in flies and other animals leads to an increase in sleep drive and a homeostatic sleep rebound [24,25,44], as well as impaired cognitive capacities such as those measured by visual learning [45,46] and attention [47]. Yet starvation-induced sleep loss seems to absolve animals from sleep need and some of the functional consequences of sleep loss [47–49]. The mechanisms supporting this surprising effect are unclear, and it is unknown what aspects of cognition are preserved under this regime. We used *upd2* mutants and tissue-specific knockdown of *upd2* and its receptor to address possible consequences of a chronic starved-like state. We demonstrate that reduced *upd2* signaling disrupts daytime sleep and leads to increased feeding-related behaviors, such as nighttime hyperphagia (increased feeding at night). While sleep deprivation typically impairs attention, we found that *upd2* knockdown animals had improved attention, even though they slept less. Finally, we show that *upd2* regulates sleep and attention via cells expressing *insulin-like peptide 2* (Ilp2) signaling in the brain. Our results highlight a role for peripheral signaling in co-regulating cognition and sleep as a function of nutrition.

## Results

### *upd2* mutants have irregular feeding and fragmented sleep

Homozygous *upd2* deletion mutants (*upd2^Δ*), which lack the 5′ UTR and the first 89 amino acids of the protein [33], have been shown to be smaller and slimmer than control animals [37]. We first measured the food intake of mutant animals to address whether the difference in their body size (Fig 1A) was due to a decrease in feeding, which would be inconsistent with a starvation cue. We used an optimized version of the capillary feeding (Café) assay [50] and tracked their food consumption over 24 hours. In agreement with previous findings, there was no significant change in total food consumption in *upd2^Δ* mutants, compared with controls (Fig 1B) [37,51]. However, when we looked at day and night feeding separately, we noticed that the mutants were mostly feeding during the night, which was opposite to the feeding rhythm of the background controls (Fig 1C). To determine if light entrainment might be driving the altered feeding behavior, we conducted the same experiment in the dark. We found that in the absence of light cues, *upd2^Δ* mutants still fed more during their subjective night,

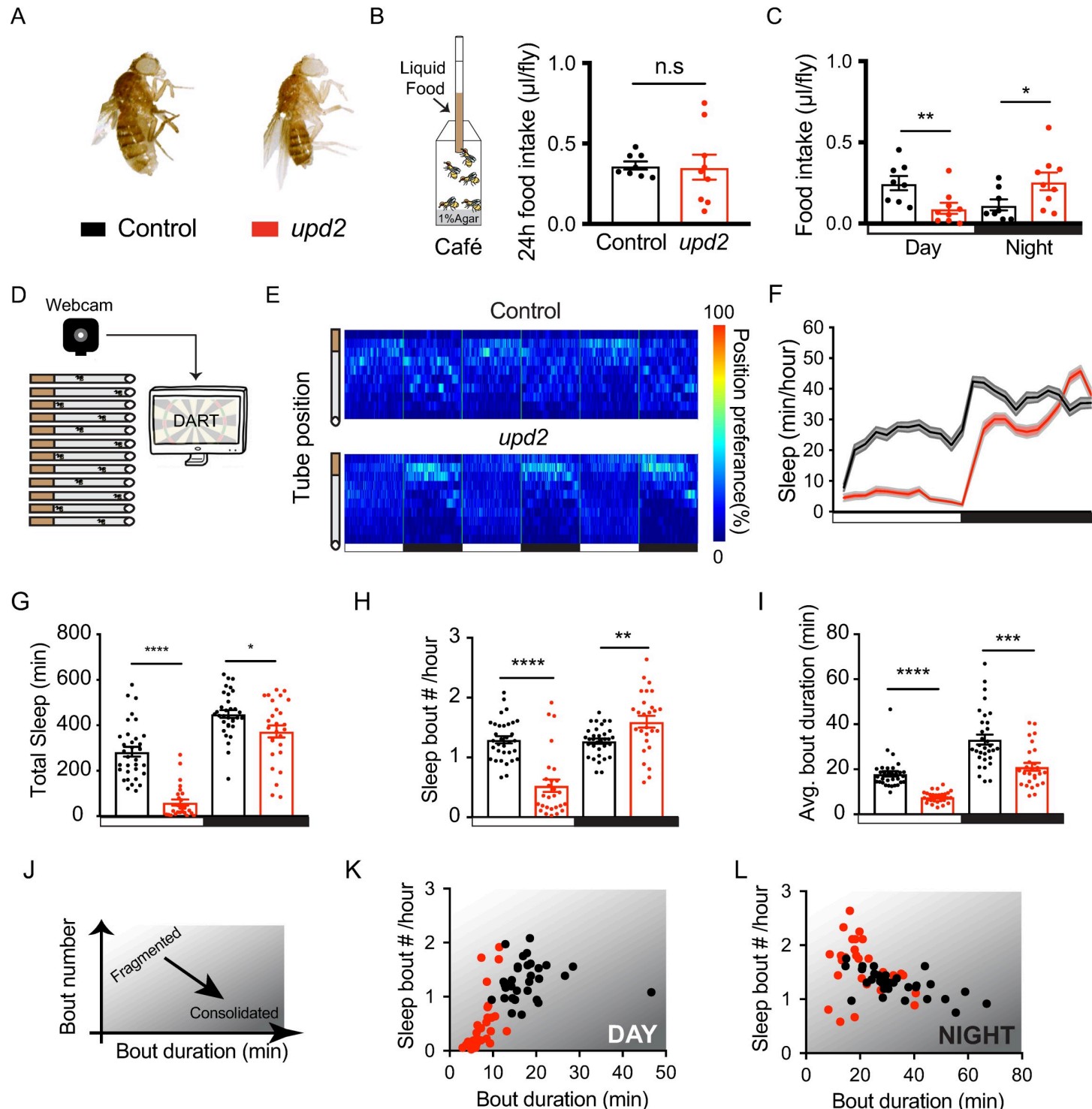

**Fig 1. *upd2* mutants have altered feeding behavior and fragmented sleep.** (A) Homozygous *upd2* deletion mutant females (right, red), compared with their background controls (left, black). (B) Food intake was measured with Café assay, with 5 flies/chamber over 24 hours (8–9 chambers, *n* = 40–45 flies per genotype). The bottom of the chamber had 1% agar to prevent desiccation. Liquid food (5% sucrose/water) was presented in a microcapillary. Total food consumption of *upd2* mutant flies (red) was similar to their background control (black). (C) Mutants had lower consumption during daytime. However, they had a significant increase in their nighttime feeding. (D) *Drosophila* arousal tracking (DART) was used to measure sleep duration in *upd2* mutants. Three- to five-day-old female virgins were placed in glass tubes and sleep was tracked over 3 days (*n* = 31–32, per genotype). (E) Average position preference heatmaps show that *upd2* mutants have an increased presence at the food site at night (black bars), whereas controls remain in the center. (F) The 24-hour sleep profile of *upd2* mutants compared with control. White and black bars represent light and dark

periods, respectively. (G) *upd2* mutants had a significant reduction in average sleep duration for both day and night. (H) The number of sleep bouts was reduced during daytime and increased during nighttime. (I) The average bout duration of *upd2* mutants was reduced for both day and night. (J) Bout number plotted against bout duration is reflective of sleep quality. Sleep is more fragmented when bout durations are short and bout numbers are high, whereas sleep is consolidated with low bout numbers and longer bout durations. (K-L) Total bout number plotted against average bout duration (minutes) showed that *upd2* mutants had fragmented day and night sleep. Student *t* test for normally distributed data or Mann-Whitney U rank-sum test for nonparametric data was used to compare data sets. $^*P < 0.05$, $^{**}P < 0.01$, $^{***}P < 0.001$; error bars show SEM. The data underlying this figure can be found in S1 Data. Café, capillary feeding; DART, *Drosophila* arousal tracking; n.s., nonsignificant; *upd2*, unpaired 2.

although feeding during their subjective day was not different from controls (S1 Fig). This shows that *upd2$^\Delta$* animals are well fed, although their feeding times seem dysregulated.

Nutritional state has been shown to influence sleep duration, as well as sleep quality [29,52]. We used the *Drosophila* arousal tracking (DART) system [53] to monitor sleep duration in *upd2$^\Delta$* flies that were housed in small glass tubes over multiple days and nights (Fig 1D). In accordance with our Café results (Fig 1C), position preferences of flies in the tubes revealed that *upd2* mutants consistently stayed near the food during the night, unlike the background controls (Fig 1E). *upd2$^\Delta$* displayed a regular day-night sleep profile (sleeping less during the day and more at night, Fig 1F); however, they slept significantly less than control flies, especially during the day (Fig 1G). Closer analysis revealed a decreased number of sleep bouts, which were shorter in duration during the day (Fig 1H and 1I). During nighttime, however, mutants had increased sleep bout numbers, which were shorter in duration (Fig 1H and 1I). These results also indicate that sleep in *upd2* mutants is more fragmented than in control animals, both during both day and night (Fig 1J, 1K and 1L). Overall, the observed decrease in sleep duration, the increased sleep fragmentation, and the misregulation of feeding suggested a maladjusted nutritional cue in these mutants, presumably resulting from the absence of a cytokine signal that normally results from adequate feeding. Closer examination of the flies' walking speed revealed that the mutants were just as active as controls (S2 Fig). Starving *upd2* mutants did not further decrease their sleep, which is already almost floored during the day (S3 Fig).

## Upd2 secretion from FBs regulates feeding and sleep quality

The *Drosophila* FB is the main tissue for energy storage, fulfilling functions similar to mammalian adipose tissue and liver [54,55]. In *Drosophila*, Upd2 is mainly secreted from the FB [33]. To test the role of adult Upd2 secretion on sleep and feeding, we used RNA interference (RNAi), expressed via a FB-specific driver line (*yolk-GAL4*), which expresses only in adult flies [56]. We measured food consumption in flies in which *upd2* had been down-regulated in the FB, specifically, to determine if this recapitulated the effects seen in *upd2* mutants. This is indeed what we found: nighttime feeding was significantly increased (hyperphagia) compared with both genetic controls (Fig 2A). Daytime feeding was not different from controls, although a trend towards overall increased feeding was noted (S4 Fig). Importantly, this shows that *upd2*-down-regulated flies are not undernourished, compared with controls. Also similar to the *upd2* mutant phenotype, FB-specific *upd2* knockdown resulted in a significant suppression of daytime sleep (Fig 2B and 2C), which was also more fragmented (S5 Fig). Together with the preceding results, this suggests that the cytokine signal affecting sleep and feeding emanates from the FB.

*upd2* is also found to be expressed in muscle tissue [33,57], so it remained possible that the cytokine was not localized to the FB. To determine whether the effect on sleep and food consumption was specific to expression in the FB, we assessed the effect of *upd2* knockdown in muscle tissue. For this, we employed a muscle-specific driver, *24B-GAL4* [58]. We found that muscle-specific knockdown of *upd2* had no effect on food consumption or sleep behavior (Fig 2D, 2E and 2F; S4 Fig). We then asked whether down-regulation of *upd2* in neuronal tissue could lead to a starved-like state. One of the other *unpaired* ligands, *upd1*, is expressed in a

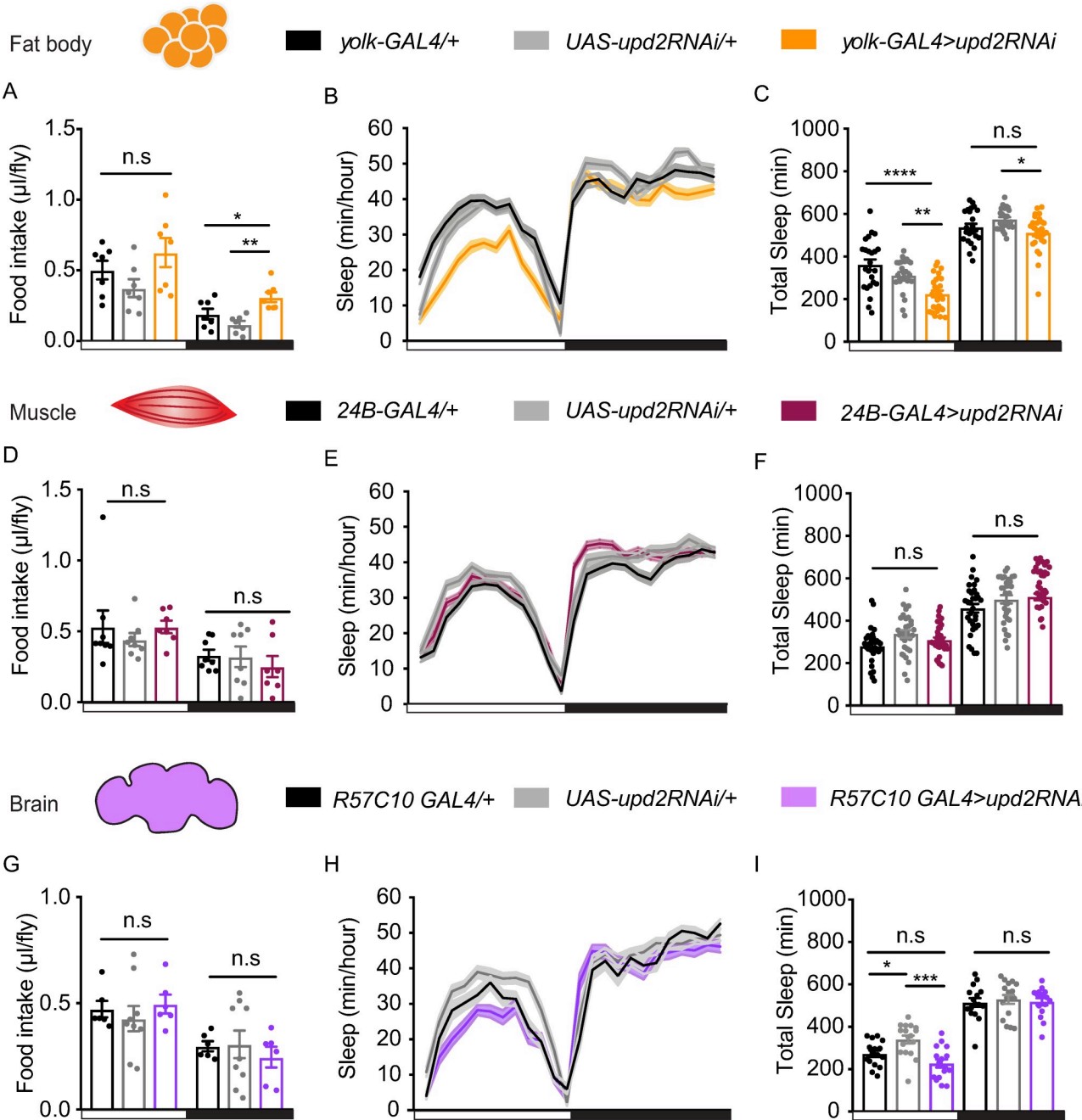

**Fig 2. *upd2* expression in the FB is required for sleep and feeding regulation.** (A) Daytime feeding (left panel, white) in flies with FB *upd2* knockdown (orange) was similar to genetic controls; *yolk-GAL4/+* (black) and *UAS-upd2RNAi/+* (gray). Nighttime feeding (right panel, black) was significantly increased in knockdown flies. (B) The 24-hour sleep profile of flies with *upd2* knockdown (orange line) compared with parental controls (black, *yolk-GAL4/+*; gray, *UAS-upd2RNAi/+*) (mean ± SEM, *n* = 24–28 per genotype). (C) Sleep duration in knockdown flies was significantly reduced during the day against both controls. Nighttime sleep was only reduced compared with *UAS-upd2RNAi/+*. (D) Muscle-specific knockdown of *upd2* by using *24B-GAL4* (maroon) did not alter day or nighttime food intake or (E-F) sleep duration. (G) Knockdown of *upd2* pan-neuronally using *R57C10-GAL4* (purple) also had no effects on food intake or (H-I) on sleep (*n* = 16–17). One-way ANOVA (with Tukey's post hoc test) for normally distributed data or Kruskal-Wallis test (with Dunn's multiple comparison test) for nonparametric data was used to compare data sets. $^*P < 0.05$, $^{**}P < 0.01$, $^{***}P < 0.001$; error bars show SEM. White and black bars indicate day and night, respectively. In A, D, and G, *n* = 5 flies per data point (chamber). The data underlying this figure can be found in S1 Data. FB, fat body; GAL4, galactose-responsive transcription factor; RNAi, RNA interference; UAS, upstream activation sequence; *upd2*, unpaired 2.

small cluster of cells in the brain [51], but it is currently unknown if *upd2* is expressed in neurons. Pan-neuronal knockdown (via *R57C10-GAL4* [59]) of *upd2* had no effect on food consumption (Fig 2G, S4 Fig). We noted significantly decreased daytime sleep compared with one of the genetic controls (Fig 2H and 2I), suggesting a possible neural source for the cytokine's effect on sleep. However, our combined data so far support the conclusion that the most robust effects of *upd2* on both sleep and feeding result from its secretion from the FB. We therefore subsequently focused on Upd2 signaling from the FB.

Decreased sleep duration does not necessarily imply decreased sleep quality; flies could be sleeping more deeply in shorter bouts and thereby still achieving key sleep functions [53,60]. We therefore next investigated sleep intensity in *upd2*-down-regulated flies. To measure sleep intensity, we delivered a series of vibration stimuli every hour and analyzed the proportion of sleeping flies responding to the stimulus (Fig 3A). We binned the flies into 10-minute sleep duration groups, which describes how long flies were asleep prior to the vibration stimulus (Fig 3A). For example, if a fly had last moved 25 minutes before the stimulus, that fly would be placed in the 21–30-minute bin, in order to determine average behavioral responsiveness (i.e., sleep intensity) for all flies in that time bin. Thus, all flies received the exact same number of stimuli, although their responsiveness varied as a function of their prior sleep duration. As in previous studies, behavioral responsiveness was qualitative, meaning that any movement above a minimum threshold was indicative of a fly having been awakened (see Materials and methods). To accurately assess sleep intensity, we ensured that all sleep duration groups had a sufficient number of arousal-probing events in all of our genetic variants (Fig 3B). Because daytime sleep bouts were rarely longer than 30 minutes, there were comparatively fewer probing events possible for longer daytime sleep bouts (Fig 3B). We found that down-regulating *upd2* in the FB resulted in significantly lighter daytime and nighttime sleep for almost all sleep durations, compared with genetic controls (Fig 3C and 3D). This suggests that lack of a cytokine signal from the FB results in overall decreased sleep intensity, in addition to decreased sleep duration and increased feeding. One interpretation of these results is that these animals sleep less so that they can look for food instead.

## Tracking sleep and feeding behavior in individual animals

In our preceding experiments, we found that *upd2* down-regulation has correlated effects on sleep and feeding behavior, although these observations were made using different assays for different sets of flies. To confirm our findings and to further our understanding of relationship between sleep and feeding, we devised a novel open-field paradigm wherein we could monitor sleep and feeding-related behaviors in the same animals. In this setup, individual flies were housed in circular arenas provisioned with standard (solid) fly food in the center of each chamber (Fig 4A). The protocol for sleep tracking was the same as in our previous experiments in small glass tubes, except flies were tracked in two dimensions (see Materials and methods). Importantly, we confirmed that the sleep profile of *upd2* knockdown flies was similar in this open-field setup, with a significant sleep reduction and fragmentation during the day (Fig 4B and 4C; S6 Fig). This shows that the reduced sleep phenotype manifests itself in different types of chambers (circular versus linear).

The circular arena setup allowed us to track the absolute location of flies in two dimensions, to show if they had any place preference. Heat plots revealed that flies frequently visited the food cup located in the center of the arena (Fig 4D), presumably to feed. We devised an automated system to quantify feeding-related behavior over multiple days (see Materials and methods), to complement sleep tracking in the same individuals. Our automated analysis was optimized following visual observation, with a fly considered to be engaged in feeding-related

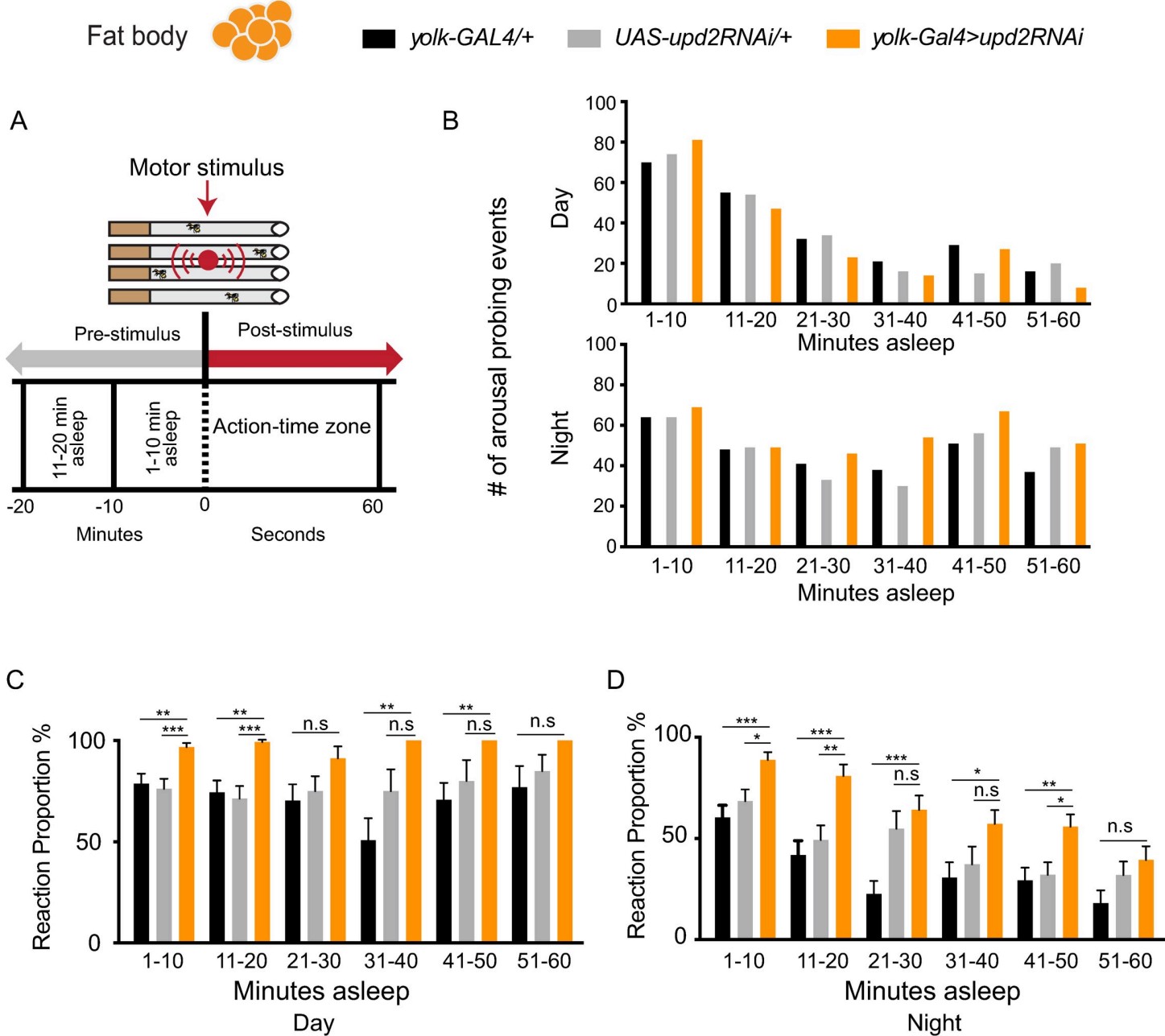

**Fig 3. *upd2* down-regulation decreases sleep intensity.** (A) A vibration stimulus was presented to flies to measure sleep intensity. A stimulus train consisted of five 0.2-second pulses presented once every hour. Flies were binned according to their pre-stimulus sleep duration (10-minute bins). Reaction proportion represents the percent of immobile animals that responded to the stimulus train (see Materials and methods). (B) The number of animals in each sleep duration bin for both day and night were similar in both genetic controls (black and gray) and in knockdown flies (orange). (C) Knockdown flies slept more lightly for most sleep durations, for both day and (D) night; n = 24–28 per genotype. Flies analyzed in this figure are from the same data set as in Fig 2A–2C. One-way ANOVA (with Tukey's post hoc test) for normally distributed data or Kruskal-Wallis test (with Dunn's multiple comparison test) for nonparametric data was used to compare data sets. $^{*}P < 0.05$, $^{**}P < 0.01$, $^{***}P < 0.001$; error bars show SEM. White and black bars indicate day and night, respectively. The data underlying this figure can be found in S1 Data. GAL4, galactose-responsive transcription factor; n.s., nonsignificant; RNAi, RNA interference; UAS, upstream activation sequence; *upd2*, unpaired 2.

behavior if it fulfilled four key criteria: (1) it was located at the food, (2) its speed at the food was less than 1 mm/second [61], (3) it remained at the food for at least 30 seconds, and (4) it was not sleeping (i.e., immobile). Together, these criteria ensured that flies were actively

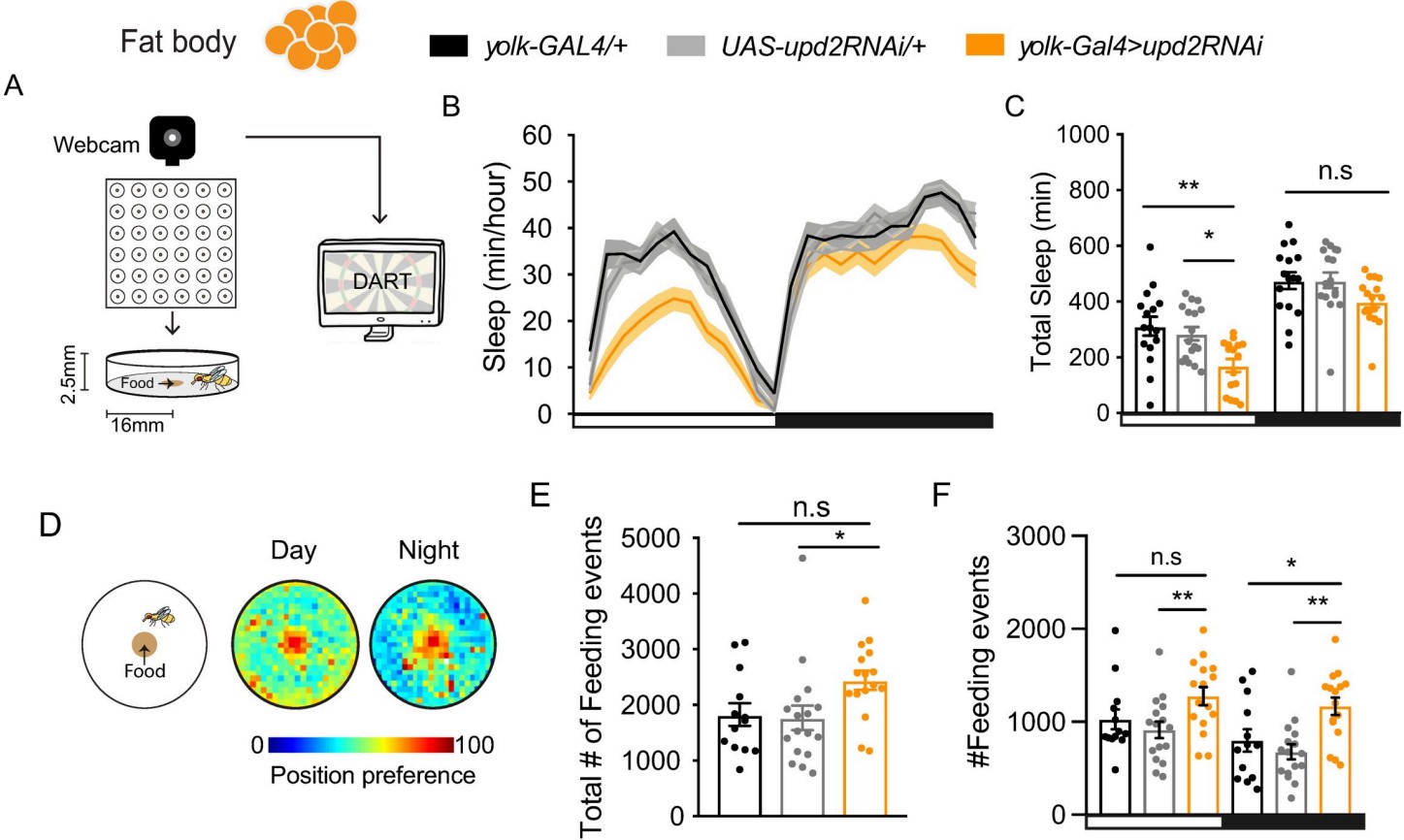

**Fig 4. Simultaneous tracking of sleep and feeding-related behavior in an open-field arena.** (A) Schema of the open-field tracking system. Flies are housed individually in circular arenas, radius = 16 mm. Each arena has a food cup in the center (radius = 5 mm). Fly activity is monitored via a webcam and analyzed with DART. (B) The 24-hour sleep profile of flies with *upd2* knockdown (orange line) compared with genetic controls. Sleep was tracked for 3 days (*n* = 15–17 per genotype) (black, *yolk-GAL4/+*; gray, *UAS-upd2RNAi/+*). (C) FB-specific *upd2* knockdown in open-field arena showed a similar phenotype as in tubes (orange bars). (D) Exemplary average 2D heatmap position preference plots for day and night, warmer colors indicate a higher probability of flies being in that position. (E) Number of feeding events for each genotype over 3 days. Total feeding events in knockdown flies was only significant compared with one of the parental controls, *UAS-upd2RNAi/+*. (F) *upd2* knockdown flies had an increase in feeding events during nighttime. Daytime feeding count was increased, as well, but was only significantly different from one of the parental controls (*UAS-upd2RNAi/+*). One-way ANOVA (with Tukey's post hoc test) for normally distributed data or Kruskal-Wallis test (with Dunn's multiple comparison test) for nonparametric data was used to compare data sets. $^{*}P < 0.05$, $^{**}P < 0.01$, $^{***}P < 0.001$; error bars show SEM. White and black bars indicate light and darkness, respectively. The data underlying this figure can be found in S1 Data. DART, *Drosophila* arousal tracking; FB, fat body; GAL4, galactose-responsive transcription factor; RNAi, RNA interference; UAS, upstream activation sequence; *upd2*, unpaired 2.

interacting with the food rather than just walking past the food or sleeping at the food. In order to confirm that we were indeed tracking feeding-related behavior, we conducted a number of additional experiments. When we prevented access to the food source by covering the food cup with parafilm, this significantly decreased the number of "feeding" visits to the center of the arena (S7A Fig). When we introduced starved flies into the arena, this doubled the number of visits to the food cup (S7B Fig). Visual inspection of fly videos taken from the above experiment confirmed that flies were indeed visiting the food cup to feed and that our behavioral criteria (e.g., the >30-second cutoff for individual feeding events) were accurate (S7C, S7D and S7E Fig). When we provided diluted food, the number of feeding events did not increase as it did with starvation, but flies slept less (S7F and S7G Fig), suggesting a nutritional deficit. Confident that we were indeed tracking feeding-related behavior, we proceeded to quantify the number of feeding events over three days and nights in our genetic variants. We observed a trend towards an increased number of feeding events in flies with *upd2* down-

regulated in the FB, compared with genetic controls (Fig 4E). When we again partitioned our analysis between day and night, we observed that *upd2* knockdown significantly increased the number of feeding events during the night (Fig 4F). Importantly, these results match closely with our Café assay results for these same genetic variants (Fig 2A), suggesting that increased feeding-related behavior in the circular arenas indicates increased food consumption. Additionally, our combined assays indicate that the feeding phenotype is unlikely to be an artifact of different assay conditions (e.g., the liquid food of the Café assay) or due to group housing in Café chambers. Thus, regardless of the assay employed, removing a cytokine signal from the FB reliably increases nighttime feeding, and therefore hyperphagia. The observed sleep reduction and sleep fragmentation resulting from *upd2* down-regulation aligns with the behavior of starved animals more generally [29]. Moreover, the hyperphagia of *upd2* mutants suggests that a signal communicating that food has been consumed is not being integrated or processed, even if flies are well fed.

## Neural correlates of starvation in the ellipsoid body R4-neurons

If *upd2* mutants are failing to process a feeding-related signal, then neural evidence of this "chronically starved" state might be evident in brain activity. Several neurons in the *Drosophila* brain reflect nutritional effects, including lateral horn leucokinin (LHLK) neurons and neuropeptide F (Npf) neurons (for a review, see [21,62]). Interestingly, certain neurons in the ellipsoid body (EB) in the central brain (which is involved in sleep as well as visual behaviors [44,63]) appear to be responsive to starvation cues: a previous study found that acute starvation increased the activity levels of R4 neurons in the EB [64]. This suggested we could utilize these neurons to determine if *upd2* knockdown animals were indeed chronically starved. We used a calcium-dependent nuclear import of LexA (CaLexA) reporter [65] to visualize activity levels in the brains of *upd2* mutants, and compared these to animals that were actually starved. We expressed the reporter construct in the R4 neurons (using *R38H02-GAL4*; S8A Fig), which labels the same neurons as in the aforementioned study [64]. We tracked both sleep and feeding behavior in individual animals in the open-field arena (as in Fig 4) for two days, after which we dissected and imaged their brains (Fig 5A). In control animals, we found that activity in the R4 neurons was negatively correlated with the number of feeding events (Fig 5B and 5C), but not with sleep duration (S8B Fig). This suggests that the R4 neurons are indicative of nutritional status in our paradigm, but not of sleep differences among individuals.

Consistent with the above data and a previous study [64], we observed a significant increase in R4 neuron activity after 24 hours of starvation (Fig 5D, first versus second panel; Fig 5E). We then placed the CaLexA/R38H02 reporter in an *upd2* mutant background to investigate R4 neuron activity in these flies. Interestingly, *upd2* mutants display increased R4 neuron activity, like starved flies, even though they had ad libitum food access and consumed amounts similar to fed controls (Fig 5D, third panel). The average activity of R4 neurons (quantified by GFP intensity) in the mutants was comparable to the level observed in starved wild-type flies (Fig 5E). Starving *upd2* mutants did not further increase R4 activity levels (Fig 5D and 5E). Overall, our results support the conclusion that lack of the Upd2 signal produces a chronically starved-like state, associated with this salient neural signature in the fly's central brain. We therefore refer to *upd2* down-regulation as a chronic starvation signal from here on.

## *upd2* down-regulation in the FB sharpens visual selective attention

Sleep-deprived flies have been shown to have deficits in learning and memory and visual attention tasks [45,47,66]. Yet starvation-induced sleep loss seems to preserve performance in some behavioral assays [49]. Indeed, many behavioral studies exploit starvation as a way to increase

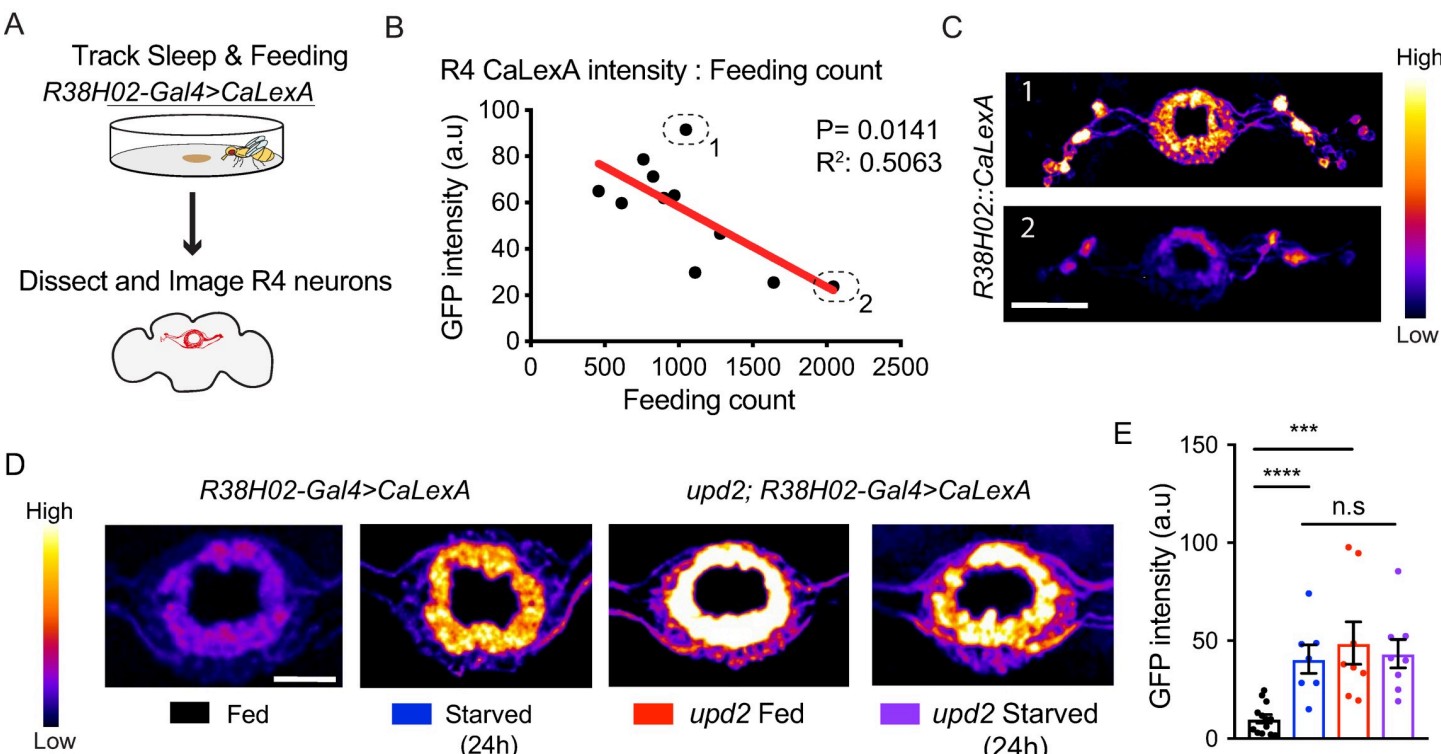

**Fig 5. *upd2* mutants show increased CaLexA expression in EB R4 neurons.** (A) Flies were housed in open-field arenas for 24 hours (starting at ZT0) for tracking sleep and feeding-related behavior. They were then collected for dissection and brains were imaged. (B) Feeding count number in open-field arenas significantly correlated with the measured CaLexA intensity of R4 neurons. (C) Sample images representing a high (number 1, upper panel) compared with a low CaLexA signal (number 2, lower panel). $n = 11$; warmer colors indicate an increase in GFP intensity. (D) Representative whole-mount brain immunostaining of fed (black) and starved wild-type (blue) (*w +; R38H02-GAL4>CaLexA*) compared with fed (red) and starved *upd2* mutant (purple) (genotype: *upd2;R38H02-GAL4>CaLexA*). Maximal intensity projections are shown in pseudo color (scale bar = 20 μm). (E) Quantification of CaLexA signals ($n = 8$–13 per condition). Control flies had ad libitum access to food from days 0 to 6. Starved group was transferred to 1% agar/starvation medium on day 5 (ZT0) until day 6 (ZT0). For correlation analysis, two-tailed *P* values for Pearson's correlation coefficient are shown. Student *t* test for normally distributed data or Mann-Whitney U rank-sum test for nonparametric data was used to compare data sets. $^*P < 0.05$, $^{**}P < 0.01$, $^{***}P < 0.001$; error bars show SEM. Flies in B and E were from a different antibody cohort. The data underlying this figure can be found in S1 Data. CaLexA, calcium-dependent nuclear import of LexA; EB, ellipsoid body; GFP, green fluorescent protein; *upd2*, unpaired 2.

motivation and to even improve performance [67–69]. Since most sleep-monitoring assays for *Drosophila* do not provide much insight into behavioral processes beyond locomotion, it remains unclear how starvation might preserve or improve behavioral performance in spite of less sleep.

We investigated whether down-regulating *upd2* affected visual selective attention. To study visual attention in flies, we used a modified version of Buridan's paradigm to track visual fixation behavior in freely walking animals (Fig 6A) [70,71]. To ensure that vision was normal in our genetic variants, we outcrossed them to *white*⁺ so that their eye pigmentation was wild-type (see Materials and methods). We then proceeded to test them first for simple visual behaviors, namely object fixation [72,73] and optomotor responsiveness [74,75]. Fixation behavior was not different from controls in *upd2* knockdown flies: animals responded normally to two opposing "target" bars by walking decisively back and forth between them (measured by their low target deviation angle, see Materials and methods) (Fig 6A). Optomotor behavior was also not significantly impacted in *upd2* knockdown animals (Fig 6B), suggesting these flies are able to perceive motion normally, along with being able to fixate on target objects. To investigate visual attention, we combined the two different kinds of visual stimuli

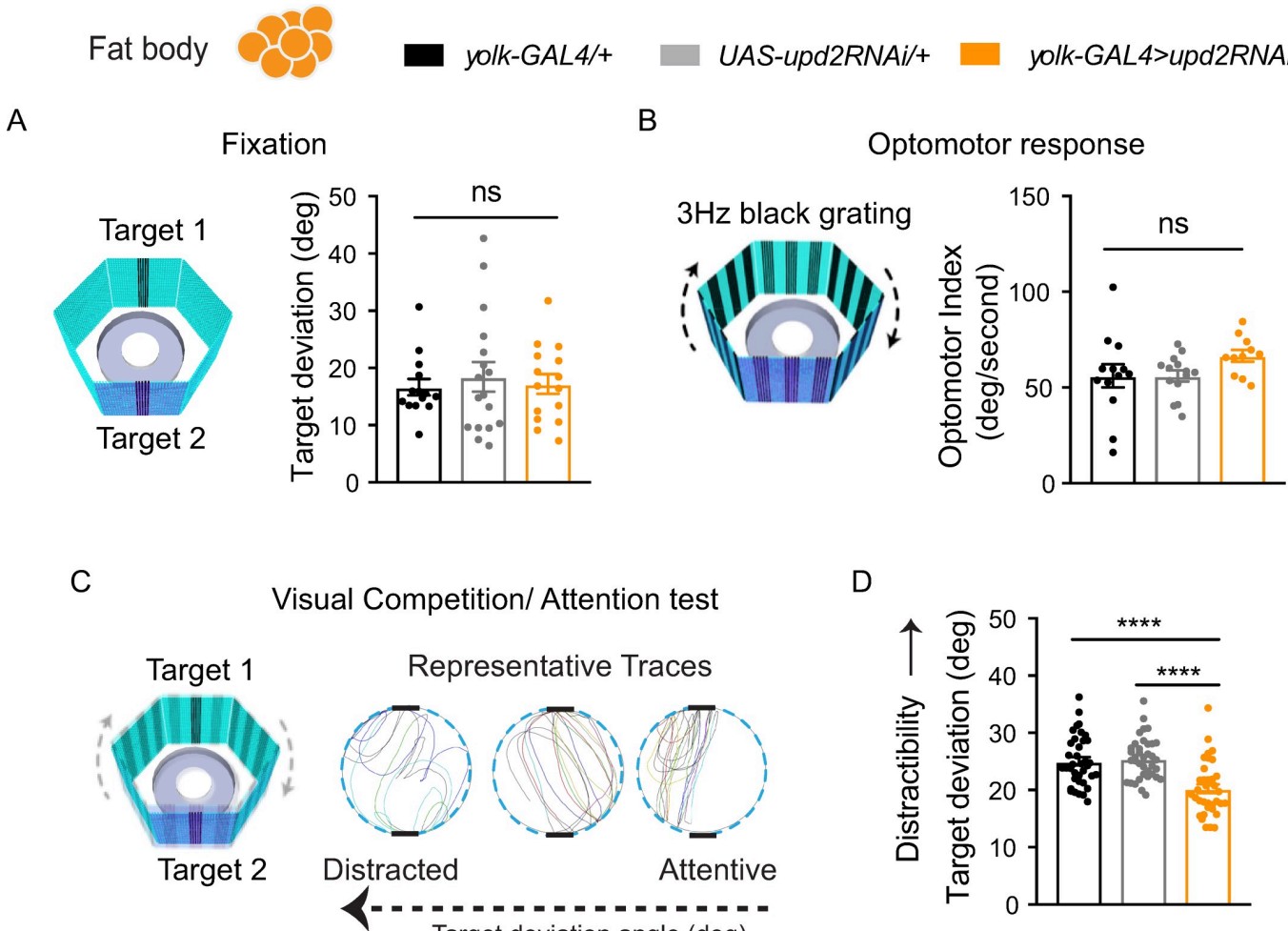

**Fig 6. Visual behavior in *upd2* deficient flies.** (A) Fixation behavior in Buridan's paradigm, in which flies were presented with two opposing black bars flickering at 7 Hz. Fixation responses were calculated by target deviation in degrees and was nonsignificant between genetic controls (gray and black bars) and knockdown flies (orange bar) (*n* = 15). (B) Optomotor response to motion stimulus (3-Hz grating) determined by the fly's turning angle (degrees per second) for genetic controls (gray and black bars) and knockdown flies (orange bar) (*n* = 11). (C) For visual competition (attention) experiments, flies were presented with the two flickering bars (fixation targets) and the 3-Hz grating (distractor) together. The target deviation angle represents a measure of distractibility. Representative traces show the trajectories of single flies with low (right), average (middle), and high (left) target deviation angles. (D) Target deviation for knockdown flies (orange bar) was significantly lower compared with genetic controls, indicating less distractibility (*n* = 21). One-way ANOVA with Tukey correction was used for comparing different conditions. *$P < 0.05$, **$P < 0.01$, ***$P < 0.001$, ****$P < 0.0001$; error bars show SEM. The data underlying this figure can be found in S1 Data. GAL4, galactose-responsive transcription factor; RNAi, RNA interference; UAS, upstream activation sequence; *upd2*, unpaired 2.

(target and optomotor) in a visual competition scenario (Fig 6C), which allowed us to measure how much the moving grating distracted flies from the target stimulus [47]. Surprisingly, in this attention paradigm, *upd2* knockdown animals performed significantly better than controls, meaning that they were less distracted by the moving grating (Fig 6D).

*upd2* mutants sleep less than control flies (Figs 1–4), but it is unclear if they are sleep deprived. Sleep-deprived wild-type flies sleep more deeply after deprivation [76], to presumably recover lost deep sleep functions. Contrary to this, *upd2* knockdown flies sleep more lightly (Fig 3). Sleep deprivation also impairs attention in flies [47], contrary to our results in *upd2* knockdown animals. This made us question if the increased visual focus observed in

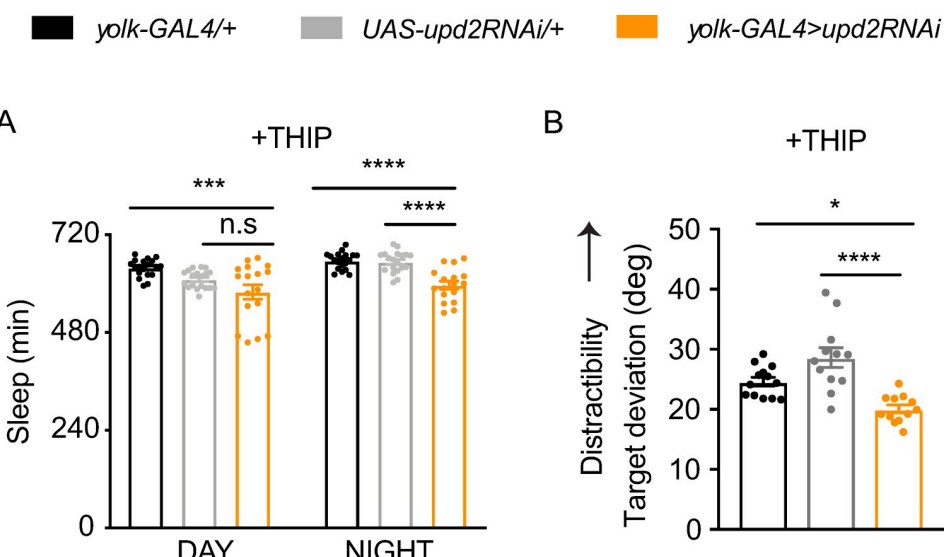

**Fig 7. Additional sleep does not restore attention in *upd2* knockdown flies.** (A) Total sleep duration in THIP-fed flies (0.1 mg/mL) was increased (compared with the same genotypes in Fig 2). *upd2* knockdown flies still showed a significant decrease in their sleep duration compared with genetic controls (*n* = 17–19). (B) Additional sleep via THIP did not rescue the improved attention phenotype of flies with FB *upd2* knockdown (orange, *yolk-GAL4>upd2RNAi*) compared with controls (*yolk-GAL4/+*, black, and *UAS-upd2RNAi/+*, gray) (*n* = 12–13). One-way ANOVA with Tukey correction was used for comparing different conditions. *P < 0.05, **P < 0.01, ***P < 0.001, ****P < 0.0001; error bars show SEM. The data underlying this figure can be found in S1 Data. FB, fat body; GAL4, galactose-responsive transcription factor; RNAi, RNA interference; UAS, upstream activation sequence *upd2*, unpaired 2.

*upd2*-deficient animals was a failure of attention, rather than improved attention. In other words, if attention is optimal in well-rested wild-type animals [77], a decreased capacity to detect a distracting stimulus (e.g., a moving grating) might be viewed as defective rather than improved attention. To address this conundrum, we decided to restore normal levels of sleep to *upd2* mutants, to see if this returned their visual attention to control levels. To increase sleep in *upd2* mutants, we used a sleep-promoting drug, THIP (4,5,6,7- tetrahydroisoxazolo-[5,4-c]pyridine-3-ol). Previous work has shown that THIP-induced sleep restores behavioral plasticity and attention to *Drosophila* mutants [28,78], so we were curious if increased sleep in *upd2* knockdown animals would return distractibility to control levels, which would argue that the increased fixation observed in these animals was a failure of attention corrected by sufficient sleep. As expected, THIP exposure increased sleep in *upd2* knockdown animals (Fig 7A, compare with Fig 2C). However, THIP exposure in *upd2* knockdown animals did not change their visual attention phenotype compared with similarly treated controls (Fig 7B). This suggests that the increased focus (or decreased distractibility) in these flies is a direct feature of their starved-like state rather than suboptimal attention processes resulting from insufficient sleep.

We showed earlier that *upd2* down-regulation in the FB decreased sleep, while down-regulation in the muscle had no effect on sleep and down-regulation in the nervous system had some effects (Fig 2). To determine whether this FB specificity extended to visual attention behavior as well, we next examined whether down-regulation of Upd2 in these other tissues (i.e., muscle and neurons) also altered visual attention. We found that *upd2* knockdown in muscle and neurons had no effect on visual behaviors or on visual attention (S9 Fig). This confirms the *Drosophila* FB as a relevant cytokine source for producing this suite of associated effects (decreased sleep, hyperphagia, and improved attention).

## Knockdown of *dome*, the *upd2* receptor in Ilp2 neurons, recapitulates *upd2* knockdown phenotypes

We next investigated how down-regulation of *upd2* might be signaling a starvation signal to the fly brain. The *upd2* receptor, *dome*, is expressed in several subsets of neurons, including mushroom body neurons, Npf neurons, and in the pars intercerebralis (PI) region of the fly brain, where Ilp neurons are located [35,51]. *Drosophila* Ilp neurons share functional similarities with mammalian insulin cells and are involved in nutrient sensing [79,80]. Ilp2, one of the three Ilps expressed in the insulin-producing cells (IPCs), (Fig 8A), has been previously shown to alter sleep and feeding [81,82]. We therefore employed an RNAi strategy to down-regulate *dome* in Ilp2 neurons (Fig 8B). As before, we investigated sleep, feeding, and visual behaviors. Remarkably, these mirrored closely all of the *upd2* knockdown phenotypes: daytime sleep in *dome* knockdown flies was decreased and more fragmented compared with genetic controls (Fig 8C and 8D; S10 Fig). *dome* knockdown flies also displayed significantly more feeding-related behavior at night (Fig 8E) and a trend to overall more feeding in general (S11 Fig). Finally, when we investigated visual attention in *dome* knockdown animals, we saw significantly improved visual attention, compared with genetic controls (Fig 8F), while there were no differences in their simple visual behaviors (S12 Fig). These results suggest that the brain integrates the peripheral Upd2 signal secreted from the FB via the Dome receptor in Ilp2 neurons, among perhaps other neurons, to simultaneously decrease sleep while sharpening visual attention.

## Discussion

Sleep has been found to be necessary for maintaining cognitive properties such as memory, attention, and decision-making in several animals, including flies and humans [17,47,78,83–86]. This is because sleep most likely accomplishes a variety of important conserved functions, for any animal brain [60,77]. It is therefore remarkable that sleep can under some circumstances be suppressed, as in the case of starvation. This suggests downstream pathways that can override some of the deleterious effects of sleep deprivation, especially effects relating to cognitive performance. That such mechanisms should exist seems adaptive: starving animals need to find food quickly, rather than sleep.

One important category of decisions animals make on a daily basis is their food choices and how to source them. In humans, insufficient sleep has been found to alter desire for food and increase the preference for high caloric foods [87]. Additionally, nutrition strongly alters sleep amount and sleep quality [88]. Poor food choices are the most common cause of metabolic disorders such as type 2 diabetes and obesity [89,90], and these disorders are also associated with sleep disorders [91,92]. Yet, how sleep and feeding decisions influence each other is not well understood, and hard to disentangle in human patients. Manipulating satiation cues in animal models provides a way to establish some level of causality that is hard to achieve by starvation per se. By manipulating a genetic starvation signal in *Drosophila*, we were able to investigate effects on sleep, feeding, and attention without the confound of actual starvation, which obviously can affect behavioral performance for a variety of spurious reasons.

A key roadblock toward addressing these questions in the *Drosophila* model has been a lack of paradigms in which sleep quality and feeding behavior can be quantified in the same animals. Studying both behaviors simultaneously has been difficult given the small size of flies and their miniscule food intake. So far, the only platform which allowed this was the Activity Recording Capillary (ARC) Feeder [52,93]. In our current study, we provide an alternative approach to measuring both behaviors by using a solid food source in open-field arenas amenable to visual tracking for sleep experiments. Our combined feeding and sleep paradigm

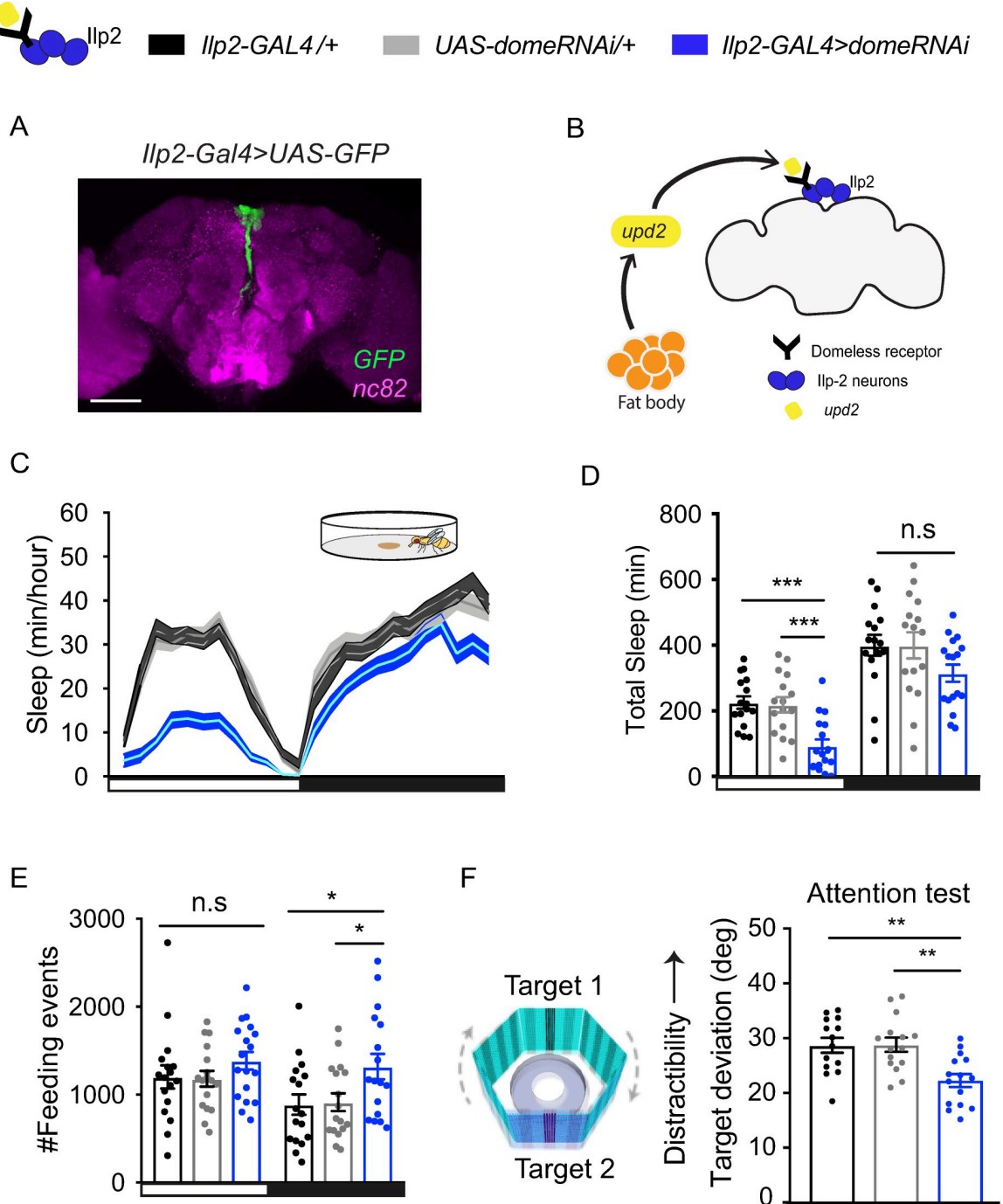

**Fig 8. Knockdown of *dome*, the *upd2* receptor, in Ilp2- expressing neurons, recapitulates *upd2* knockdown phenotypes.** (A) *Ilp2-GAL4* neurons are located in the PI region of the brain. Whole-mount brain image, maximum projection, scale bar = 100 μm. (B) Schema illustrating the integration of *upd2* signal via *dome* receptor in Ilp2 neurons. (C) The 24-hour sleep profile of flies with knockdown of *dome* receptor in Ilp2 neurons (blue) compared with genetic controls (*Ilp2-GAL4/+*, black; *UAS-domeRNAi/+*, light gray). Sleep was measured over 3 days in open-field arenas. (D) Daytime sleep was significantly reduced in knockdown flies, whereas nighttime sleep was not affected. (E) The number of feeding events showed a significant increase at nighttime. (F) Target deviation angle of flies with *dome* knockdown was significantly lower compared with genetic controls, indicating less distractibility (*n* = 13–15). For sleep analysis, 3–5-day-old adult females virgins were used, *n* = 15–17 per genotype. $^*P < 0.05$, $^{**}P < 0.01$, $^{***}P < 0.001$; one-way ANOVA, error bars show SEM. The data underlying this figure can be found in S1 Data. *dome*, *domeless*; GAL4, galactose-responsive transcription factor; GFP, green fluorescent protein; Ilp2, insulin-like peptide 2; PI, pars intercerebralis; RNAi, RNA interference; UAS, upstream activation sequence; *upd2*, unpaired 2.

confirmed a role for *upd2* in regulating feeding and sleep behavior in *Drosophila*. In future work, similar combined platforms (e.g., FlyPad [94]) could be employed to provide a more precise assessment of feeding alongside sleep monitoring. This is important, as there is clearly substantial individual variability in feeding and sleep behavior (Fig 5B, S7 Fig). Our results nevertheless showed a close correspondence between feeding related behavior and food ingestion. However, feeding-related behavior and actual food consumption may not always be entirely correlated. Increased feeding-related behavior in well-fed flies may be more suggestive of a missing satiety cue than increased consumption per se, because of a likely ceiling effect on how much food flies can consume.

Satiety requires signals from peripheral tissues to terminate food intake. The FBs are the main energy storage site in flies and fulfill functions similar to mammalian adipose tissue and liver [54]. In *Drosophila*, *upd2* is secreted from the FBs upon feeding [37]. Similarly, the mammalian ortholog leptin is secreted by adipose tissue and is received as a satiety signal by the brain to inhibit appetite [38,95]. Leptin levels in the blood change upon food intake [96] and are shown to fluctuate in a circadian manner [97]. Interestingly, a number of studies have found that leptin levels in rodents are significantly decreased in response to sleep deprivation [98–101]. On the other hand, leptin-deficient mice (Lepob/ob) have been found to have altered sleep architecture, with increased non-rapid eye movement (NREM) sleep and shorter sleep bouts [102]. These studies suggest a homeostatic relationship between leptin levels and sleep amount or quality. Our study suggests that Upd2 in flies plays a similar role to leptin in the simultaneous regulation of feeding and sleep.

It is, however, unclear if *upd2* mutants need more sleep. It seems likely that key sleep functions are not being satisfied in *upd2* mutants, such as functions that might be associated with deeper sleep, for example. Nevertheless, these animals have improved visual attention, and imposing more sleep does not change their sharpened attention phenotype. One possibility is that *upd2* knockdown promotes a different kind of sleep that optimizes attention specifically. Alternatively, the satiety cue provided by *upd2* might be directly modulating attention-like behavior, irrespective of parallel effects on sleep. A direct effect of *upd2* on improving attention might ensure optimal cognitive performance in the face of suppressed sleep functions. In other words, if sleep is going to be sacrificed in order to promote foraging and feeding, then this behavior should be optimized rather than degraded. Although we did not examine foraging behavior in our study, our visual attention experiments probe a fundamental cognitive process (distractibility) that could affect many different kinds of behaviors, including foraging. Interestingly, we found that simple visual behaviors remained unaffected in *upd2* or *dome* knockdown animals. Similarly, sleep deprivation does not seem to affect visual fixation and optomotor behavior in *Drosophila* [47]. One interpretation of our results is that the decreased level of distractibility in the knockdown animals reflects improved attention, which might be an adaptation for finding appropriate food resources under suboptimal nourishment conditions. While we have not completely excluded the possibility that this could be a form of impaired attention (to be less distractible can be maladaptive, as in autism), such a sharpened focus on innately attractive objects (for a walking fly, a dark bar is attractive [26]) could be seen as beneficial when animals are foraging for food.

Along with attention and sleep, another behavior that is significantly altered in *upd2* mutants as well as knockdown animals is feeding itself, with nighttime hyperphagia being a recurrent observation using two different assays. Does decreased sleep quality alter feeding (and thus *upd2* levels), or do altered feeding patterns affect sleep quality? It remains difficult to disambiguate causality in this regard in humans, and even in animal models. In our *Drosophila* experiments, we found an increase in R4 neuronal activity in *upd2* mutants, which we propose reflects their nutritional status, as these neurons have been shown to respond under starvation

regimes [103]. This suggests that a persistent hunger signal, reflected in R4 neuronal activity, may underlie the decreased sleep phenotype. Considering the similarities between leptin/upd2 regulation, future studies temporally manipulating this hunger cue at different points in the signaling pathway should be able to disambiguate causal links between sleep need and nutritional status. It will also be of interest in future research to determine if there is a long-term cost to sacrificing sleep by down-regulating satiety cues to the brain.

Our findings suggest that IIp2 cells are a key portal in the fly brain for integrating hunger signals and translating these into appropriate behavioral programs. How signaling from IIp2 cells in the fly brain leads to improved visual attention remains unknown. In fly larvae, Ilp2 is released within the brain and acts on a subset of neurons (e.g., Hugin neurons [104]) via insulin receptors. So it is possible that, in adults, insulin-expressing cells such as Ilp2 may target the EB indirectly via insulin receptors in the EB, and that this would regulate selective attention by tuning circuits in the central complex that affect visual behavior more broadly [105,106]. Furthermore, signals downstream of the insulin receptors have been shown to regulate gene expression, such as for other receptors [107,108], so insulin signaling from the Ilp2 (or other Ilp) cells might have far-reaching effects on neuronal functions. Alternatively, insulin-expressing cells might be communicating more directly to arousal-regulatory circuits in the central complex, by way of gap junctions, for example [109], to directly modulate behavioral responsiveness levels. Future experiments testing either of those possibilities should reveal the downstream mechanisms involved.

## Materials and methods

### Fly stocks and maintenance

Flies were raised on standard agar-yeast-based food at 25˚C, 50%–60% humidity with 12-hour light: 12-hour dark cycle. Starvation experiments were performed on 1% agar food for 24 hours. *upd2$^{\Delta 3-62}$* was kindly provided by J. Hombria [33]. The following stocks were from Bloomington Drosophila Stock Center (BDSC), *yw* (#1495), *yolk-GAL4* (#58814), *UAS-upd2RNAi* (#33988, HMS00901) [110], *UAS-domeRNAi* (#32860), *Ilp2*-Gal4 (#37516), *24B-GAL4* (#1757), *R57C10-GAL4* (#39171), *R38H02-GAL4* (#47352) [111], *UAS-CaLexa* (#66542) [65]. Flies were crossed into a *w+* (*Canton-S*) background (lab stock, originally obtained from BDSC) for all knockdown experiments.

### Sleep and sleep intensity measurements

DART software was used for sleep tracking and analysis [53]. Three-to-five-day-old virgin females were placed into 65-mm glass tubes (Trikinetics, Waltham, MA). A Logitech webcam (c9000 or c920) was used for sleep recordings with 5 frames per second. Sleep parameters were calculated according to traditional fly sleep criteria (Shaw and colleagues, 2000; Hendricks and colleagues, 2000), where sleep is defined as inactive durations for 5 minutes or more (a "sleep bout"). These were binned for every hour ("sleep minutes/hour"). For sleep intensity measurements, a vibration stimulus (a train of five 200-ms pulses set at 3.0 G) using motors (Precision Microdrives, 312–101) was presented every hour to measure behavioral responsiveness. Details of the software and calculations are described in Faville and colleagues, 2015. Behavioral responsiveness was registered if a fly moved 3 mm or more within 60 seconds of the vibration stimulus, with any movement within that time frame identified as an awakening. Responsiveness data were binned into 10-minute prior immobility epochs, depending on when flies had last shown any movement since the vibration stimulus, from 0 to 60 minutes.

## Open-field behavioral analysis

The platform was custom made with white acrylic sheet. Each platform consisted of 36 individual chambers 36 mm in diameter and 2.5 mm in height. The center of the chamber had a hole with a 3-mm depth and 5-mm width, where food was placed. The platform was covered with a transparent acrylic sheet.

Each food area was pre-filled with a layer of 1% agar to maintain the moisture of the food. Once solidified, it was covered with a layer of regular fly food.

Flies were transferred without anesthesia by using a mouth aspirator and acclimatized for minimum of 12 hours prior to experiment. Fly sleep tracking and analyses were performed using DART [53] with custom-made MATLAB (Mathworks Natick, MA) scripts. Kinematic calculations were performed as previously described [112]. For feeding analysis, the food area was detected using the Matlab function "imfindcircles" (object polarity was set to Dark). A fly was considered as feeding if it fulfilled four criteria: (1) distance from the feeding region (0 mm from food pit), (2) speed $\leq 1$ mm/second, (3) time spent feeding >30 seconds, and (4) flies were not sleeping (see above). The number of feeding events represent the total number of events throughout the recording.

## Café assay

The assay was slightly modified from the previously published versions (Ja and colleagues, 2007). Virgin female flies (6–8 days old) were used. Every testing chamber had 1% agar on the bottom, to eliminate the possibility of desiccation. Food (5% w/v sucrose, Sigma Aldrich) was presented in 5-μL micropipettes (VWR, Westchester, PA) and the level of the meniscus was measured over time. For each condition, 6–8 chambers were set, with 4–5 flies in each chamber. The experiments for different conditions were performed on the same day starting at ZT0-1, in an incubator with 25˚C and 50%–55% relative humidity. At least 5 empty chambers without flies were used to control for the effects of evaporation.

## Visual paradigm

A modified version of Buridan's paradigm was used to assay for visual attention [47]. Visual cues were presented on light-emitting diode (LED) panels. Each LED panel contained 1,024 individual LED units (32 rows by 32 columns) and was controlled via an LED Studio software (Shenzen Sinorad, Medical Electronics, Shenzen, China). Visual stimuli were created in Vision Egg software [113], written in Python programming language (L. Kirszenblat and Y. Zhou).

Three different visual cues were tested.

1. A moving grating (3 Hz) for testing the optomotor response behavior.

2. Two opposing flickering bars (7 Hz) for testing fixation behavior.

3. Competition stimulus (figure-ground), with both grating and opposing flickering bars for testing selective visual attention.

Fixation and optomotor experiments lasted 1 minute. Figure-ground experiments lasted 3 minutes, during which the grating (clockwise or counterclockwise) was switched after 1.5 minutes.

For each test, female flies were collected as virgins and kept in vials in groups of 15–20 per vial. On day 2, their wings were cut under $CO_2$ anesthesia and they were placed into fresh vials. They were given 2 days to recover from the effects of $CO_2$ anesthesia. Tests were performed on a round platform (R = 86 mm) surrounded by a water-filled moat to prevent escape. The visual stimuli were alternated between each experiment from being presented on the

horizontal or the vertical axes. Optomotor experiments were alternated between clockwise and counterclockwise gratings (1.5 minutes each). LED panels formed a hexagon, surrounding the platform (29-cm diameter, 16-cm height). The dark bar was 9 degrees in width and 45 degrees in height from the center of the arena. A camera (SONY Hi Resolution Colour Video Camera CCD-IRIS SSC-374) placed above the arena was used for tracking the movement of the fly on the platform at 30 frames per second. The open-source tracking software was used to record the position of the fly (Colomb and colleagues, 2012)

Visual responses were analyzed by using CeTran (3.4) software (Colomb and colleagues, 2012) and custom-made scripts in R programming language (L. Kirszenblat and Y. Zhou). Target deviation was calculated as the smallest angle between the fly's trajectory and either of the vertical stripes (Colomb and colleagues, 2012). Optomotor index was calculated as the angular velocity (turning angle/second) in the direction of the moving grating.

## Pharmacology

THIP, also known as gaboxadol, was dissolved in standard food at 0.1 mg/mL for two days. For sleep experiments, flies were transferred to tubes containing THIP-laced regular food. For visual behavior experiments, flies were transferred to regular food 1 hour prior to testing, as described previously [78].

## Immunohistochemistry and confocal Imaging

Flies were collected under $CO_2$ anesthesia and transferred to a drop of 1× PBS for dissection. After dissection, brains were transferred to a mini PCR-tube with 200 μL of 1× PBS. All of the following steps were performed on a rotator with 27 rpm at room temperature. Brains were fixed with 4% paraformaldehyde diluted in PBS-T (1× PBS, 0.2 Triton-X 100) for 20–30 minutes, followed by 3 washes in PBS-T. They were then blocked with 10% goat serum (Sigma Aldrich, St. Louis, MO) for 1 hour, followed by overnight primary antibody incubation. On the second day, primary antibody was removed and brains were washed 3× with PBS-T. Then the secondary antibody was added and the tube was covered with aluminum foil for overnight incubation. On day 3, secondary antibody was removed and brains were washed with PBS-T. Primary antibodies were rabbit anti-GFP 1:1,000 (Invitrogen), mouse anti-nc82 1:10 (DSHB). Secondary antibodies were anti-rabbit 488, 1:250 (Invitrogen), anti-mouse 647, 1:250 (Invitrogen). Brains were then transferred to microscope slides and mounted on a drop of Vectashield (Vector Laboratories, Burlingame, CA) for imaging. Images were acquired on a spinning-disk confocal system (Marianas; 3I) consisting of an Axio Observer Z1 (Carl Zeiss) equipped with a CSU-W1 spinning-disk head (Yokogawa Corporation of America), ORCA-Flash4.0 v2 sCMOS camera (Hamamatsu Photonics), 20× 0.8 NA PlanApo, and 100× 1.4 NA PlanApo objectives were used and image acquisition was performed using SlideBook 6.0 (3I).

For CaLexA experiments, the same acquisition settings were used between different conditions. Fiji (ImageJ) was used for image processing. GFP intensity measurements were done using the Fiji intensity measurement plug-in.

## Statistical analyses

Statistical analyses were performed using Prism 7.0a (GraphPad Software). Normality tests were performed using Shapiro-Wilk normality tests. For normally distributed data, two-tailed, unpaired Student *t* test or one-way ANOVA followed by Tukey correction was performed. Unless otherwise stated, all data sets represent mean ± SEM.

## Supporting information

**S1 Fig. Nighttime hyperphagia of *upd2* mutants is not dependent on light entrainment.**
(A) Total food intake for feeding experiments under constant darkness (DD) was not significantly different between control and *upd2* mutants. (B) *upd2* mutants (red) had decreased nighttime food intake compared to controls (black); daytime food intake was similar to controls ($n$ = 25–35 flies with 5 flies per Café chamber). *$P < 0.05$, **$P < 0.01$, ***$P < 0.001$, Student $t$ test, error bars show SEM. The data underlying this figure can be found in S1 Data. Café, capillary feeding; *upd2*, unpaired 2.
(TIF)

**S2 Fig. *upd2* mutants are more active but have comparable walking speed to controls.** (A) Average speed of *upd2* mutant flies (red) was not significantly different from controls (black). (B) Mutant flies had increased wake duration compared to controls. *$P < 0.05$, **$P < 0.01$, ***$P < 0.001$; flies in this figure are from the same data set as in Fig 1. Student $t$ test for normally distributed data or Mann-Whitney U rank-sum test for nonparametric data was used to compare data sets. *$P < 0.05$, **$P < 0.01$, ***$P < 0.001$; error bars show SEM. The data underlying this figure can be found in S1 Data. *upd2*, unpaired 2.
(TIF)

**S3 Fig. Starvation does not change sleep phenotype of *upd2* mutants.** (A) Flies were kept on regular food from days 0–3. On day 3 at ZT12, they were placed into tubes with either regular food or starvation media for sleep tracking. Recording was started at nighttime and followed for 24 hours. (B) Both fed (red) and starved (pink) *upd2* mutants slept significantly less during both night and (C) day compared to fed (black) and starved (blue) controls. $n$ = 14–17, Student $t$ test, *$P < 0.05$, **$P < 0.01$, ***$P < 0.001$, ****$P < 0.0001$; error bars show SEM. The data underlying this figure can be found in S1 Data. *upd2*, unpaired 2.
(TIF)

**S4 Fig. Total food intake of flies with *upd2* knockdown.** (A) FB knockdown of *upd2* significantly increases food intake compared with UAS-*upd2*RNAi/+. (B,C) Muscle and pan-neuronal knockdown of *upd2* shows similar food intake compared with both genetic controls. These data sets are the same as in Fig 2A, 2D and 2G. One-way ANOVA with Tukey correction was used for comparing different conditions. *$P < 0.05$, **$P < 0.01$, ***$P < 0.001$, ****$P < 0.0001$; error bars show SEM. The data underlying this figure can be found in S1 Data. FB, fat body; RNAi, RNA interference; UAS, upstream activation sequence; *upd2*, unpaired 2.
(TIF)

**S5 Fig. Sleep fragmentation in flies with FB-specific *upd2* knockdown.** (A) Bout number plotted against average bout duration (minutes) showed that *upd2* knockdown flies had fragmented day sleep. (B) Nighttime pattern was similar to controls (black, *yolk-GAL4/+*; gray, *UAS-upd2RNAi/+*; orange, *yolk-GAL4>upd2RNAi*). $n$ = 24–28 per genotype; data set plotted here is the same as in Fig 2B and 2C. The data underlying this figure can be found in S1 Data. FB, fat body; GAL4, galactose-responsive transcription factor; RNAi, RNA interference; UAS, upstream activation sequence *upd2*, unpaired 2.
(TIF)

**S6 Fig. Sleep fragmentation in open-field arena for flies with FB-specific *upd2* knockdown.** (A) Bout number plotted against average bout duration (minutes) showed a fragmentation pattern for daytime in open-field arena for *upd2* knockdown flies. (B) Nighttime pattern was similar to controls. Sleep was tracked for 3 days ($n$ = 15–17 per genotype) (black, *yolk-GAL4/+*; gray, *UAS-upd2RNAi/+*; orange, *yolk-GAL4>upd2RNAi*). Data set plotted here is the same as

in Fig 4. The data underlying this figure can be found in S1 Data. FB, fat body; GAL4, galactose-responsive transcription factor; RNAi, RNA interference; UAS, upstream activation sequence; *upd2*, unpaired 2.
(TIF)

**S7 Fig. Feeding-related behavior in open-field arena under different conditions.** (A) Number of feeding events with access to the food cup (red) compared with "no food" condition (black), where the food cup was covered with parafilm to prevent access. Exemplary heatmaps for flies under different food conditions (right panel). (B) Flies starved for 48 hours (blue bar) displayed a significant increase in feeding counts compared with control flies that had been fed (black) ($n = 13$–$15$). (C) Representative images from the video recordings. The bottom image is showing a fly feeding. (D) Visual annotation of the number of food visits displayed a significant increase in starved flies. (E) The average duration of food visit per fly was not significantly different between control and starved flies. (F) Flies on diluted food (red) (20% of regular food calories) displayed no change in their feeding counts compared with flies on regular food (black) but (G) slept less during both day and night ($n = 10$–$13$, Student *t* test for normally distributed data or Mann-Whitney U rank-sum test for nonparametric data was used to compare data sets. $^*P < 0.05$, $^{**}P < 0.01$, $^{***}P < 0.001$; error bars show SEM. The data underlying this figure can be found in S1 Data.
(TIF)

**S8 Fig. R4 neuron activity does not correlate with sleep duration.** (A) Expression of *R38H02-GAL4* in the brain using *UAS-mCD8::GFP* (green). Neuropil is stained with bruchpilot (nc82, magenta). Scale bar, 100 μm. (B) CaLexA intensity of individual flies plotted against their total sleep duration (over 24 hours). Flies were housed in open-field arenas. Two-tailed *P* values for Pearson's correlation coefficient are shown. Analyses in this figure is from the same data set as in Fig 5. The data underlying this figure can be found in S1 Data. CaLexA, calcium-dependent nuclear import of LexA; GAL4, galactose-responsive transcription factor; GFP, green fluorescent protein; UAS, upstream activation sequence
(TIF)

**S9 Fig. Knockdown of *upd2* in muscles or neurons has no effect on visual behaviors.** (A) We did not observe any differences in simple visual behaviors (fixation [$n = 15$–$20$], optomotor [$n = 6$–$8$] or in visual attention [$n = 13$–$16$] with muscle-specific *upd2* knockdown (*24B-GAL4>upd2RNAi*, maroon) compared with controls (*24B-GAL4/+*, black and *UAS-upd2RNAi/+*, gray). (B) Pan-neuronal knockdown of *upd2* (*R57C10-GAL4>upd2RNAi*, purple) also had no impact on visual behaviors (optomotor, $n = 12$–$16$), fixation, $n = 8$–$16$), and visual attention ($n = 14$–$16$) compared with genetic controls (*R57C10-GAL4/+*, black and *UAS-upd2RNAi*, gray). One-way ANOVA with Tukey correction was used for comparing different conditions. $^*P < 0.05$, $^{**}P < 0.01$, $^{***}P < 0.001$, $^{****}P < 0.0001$; error bars show SEM. The data underlying this figure can be found in S1 Data. GAL4, galactose-responsive transcription factor; RNAi, RNA interference; UAS, upstream activation sequence; *upd2*, unpaired 2.
(TIF)

**S10 Fig. Sleep fragmentation in open-field arena for flies with *dome* knockdown in *Ilp2* expressing neurons.** (A) Bout number plotted against average bout duration (minutes) showed a fragmentation pattern for daytime. (B) There was no obvious fragmentation pattern for nighttime. Flies in this figure are from the same data set as in Fig 8D and 8E. Sleep was tracked for 3 days ($n = 15$–$17$ per genotype). (*Ilp2*-GAL4/+, black; UAS-*dome*RNAi/+, light gray; blue *Ilp2*-GAL4>*dome*RNAi. $n = 15$–$17$ per genotype. Sleep was recorded over 3 days. The data underlying this figure can be found in S1 Data. *dome*, *domeless*; GAL4, galactose-

responsive transcription factor; Ilp2, insulin-like peptide 2; RNAi, RNA interference; UAS, upstream activation sequence.
(TIF)

**S11 Fig. *dome* knockdown shows a trend to increased feeding.** (A) Number of feeding events was not significantly different between control flies and *dome* knockdown flies. Flies in this figure are from the same data set as in Fig 8D and 8E. (B) Total food intake over 24 hours in Café chamber of *dome* knockdown flies was significantly increased compared to one of the genetic controls ($n = 25$, with 5 flies per chamber). One-way ANOVA with Tukey correction was used for comparing different conditions. $^{*}P < 0.05$, $^{**}P < 0.01$, $^{***}P < 0.001$, $^{****}P < 0.0001$; error bars show SEM. The data underlying this figure can be found in S1 Data. Café, capillary feeding; *dome*, *domeless*.
(TIF)

**S12 Fig. *dome* knockdown does not alter simple visual behaviors.** (A) Fixation and (B) optomotor behavior of *Ilp2*-GAL4>*dome*RNAi (blue) were not significantly different from *Ilp2*-GAL4/+, black, and UAS-*dome*RNAi/+, gray. $n = 6$–8 per experiment; one-way ANOVA with Tukey correction was used for comparing different conditions. $^{*}P < 0.05$, $^{**}P < 0.01$, $^{***}P < 0.001$, $^{****}P < 0.0001$; error bars show SEM. The data underlying this figure can be found in S1 Data. *dome*, *domeless*; GAL4, galactose-responsive transcription factor; Ilp2, insulin-like peptide 2; RNAi, RNA interference; UAS, upstream activation sequence.
(TIF)

**S1 Data. Excel spreadsheet with data listed for all main and supplementary figures.**
(XLSX)

## Acknowledgments

We would like to thank the Goetz lab (Queensland Brain Institute) for antibodies. Imaging was performed at the Queensland Brain Institute's Advanced Microscopy Facility using Yokogawa spinning disk confocal. We thank Burczyk/Faville/Kottler (BFK) for adjustments made to the DART software for tracking feeding behavior.

## Author Contributions

**Conceptualization:** Deniz Ertekin, Bruno van Swinderen.

**Data curation:** Deniz Ertekin, Leonie Kirszenblat.

**Formal analysis:** Deniz Ertekin, Leonie Kirszenblat.

**Funding acquisition:** Bruno van Swinderen.

**Investigation:** Deniz Ertekin, Leonie Kirszenblat.

**Methodology:** Deniz Ertekin, Leonie Kirszenblat.

**Project administration:** Bruno van Swinderen.

**Resources:** Deniz Ertekin, Leonie Kirszenblat, Bruno van Swinderen.

**Software:** Richard Faville.

**Supervision:** Bruno van Swinderen.

**Validation:** Deniz Ertekin.

**Visualization:** Deniz Ertekin, Leonie Kirszenblat, Bruno van Swinderen.

**Writing – original draft:** Deniz Ertekin, Bruno van Swinderen.

**Writing – review & editing:** Deniz Ertekin, Leonie Kirszenblat, Bruno van Swinderen.

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
