## [Editor Report · Decision Letter 0]

11 Oct 2019

Dear Dr van Swinderen, 

Thank you for submitting your manuscript entitled "­­Downregulation of a satiety signal from peripheral fat bodies improves visual attention while reducing sleep need in Drosophila." for consideration as a Research Article by PLOS Biology.

Your manuscript has now been evaluated by the PLOS Biology editorial staff as well as by an academic editor with relevant expertise and I am writing to let you know that we would like to send your submission out for external peer review.

Please re-submit your manuscript within two working days, i.e. by Oct 15 2019 11:59PM.

Kind regards,

Ines

--

Ines Alvarez-Garcia, PhD

Senior Editor

PLOS Biology

Carlyle House, Carlyle Road

Cambridge, CB4 3DN

+44 1223–442810

---

## [Decision Letter · Decision Letter 1]

12 Nov 2019

Dear Dr van Swinderen,

Thank you very much for submitting your manuscript "­­Downregulation of a satiety signal from peripheral fat bodies improves visual attention while reducing sleep need in Drosophila" for consideration as a Research Article at PLOS Biology. Your manuscript has been evaluated by the PLOS Biology editors, an Academic Editor with relevant expertise, and by four independent reviewers.

The reviews of your manuscript are appended below. As you will see, the reviewers find the work novel and potentially interesting, however they also raise several concerns that need to be addressed. They mention a few issues with the experimental design and interpretation, the fact that you are measuring food proximity, not feeding behaviour, and that more sophisticate analyses are needed to support the conclusions.

Following discussion with the Academic Editor about the reviews, I regret that we cannot accept the current version of the manuscript for publication. We remain, however, very interested in your study and we would be willing to consider resubmission of a comprehensively revised version that thoroughly addresses all the reviewers' comments. Please note that we cannot make any decision about publication until we have seen the revised manuscript and your response to the reviewers' comments. Your revised manuscript would be sent for further evaluation by the reviewers.

We appreciate that these requests represent a great deal of extra work, and we are willing to relax our standard revision time to allow you six months to revise your manuscript. Please email us (plosbiology@plos.org) to discuss this if you have any questions or concerns, or think that you would need longer than this. At this stage, your manuscript remains formally under active consideration at our journal; please notify us by email if you do not wish to submit a revision and instead wish to pursue publication elsewhere, so that we may end consideration of the manuscript at PLOS Biology.

Your revisions should address the specific points made by each reviewer. Please submit a file detailing your responses to the editorial requests and a point-by-point response to all of the reviewers' comments that indicates the changes you have made to the manuscript. In addition to a clean copy of the manuscript, please upload a 'track-changes' version of your manuscript that specifies the edits made. This should be uploaded as a "Related" file type. You should also cite any additional relevant literature that has been published since the original submission and mention any additional citations in your response. 

Before you revise your manuscript, please review the following PLOS policy and formatting requirements checklist PDF: http://journals.plos.org/plosbiology/s/file?id=9411/plos-biology-formatting-checklist.pdf. It is helpful if you format your revision according to our requirements - should your paper subsequently be accepted, this will save time at the acceptance stage.

Please note that as a condition of publication PLOS' data policy (http://journals.plos.org/plosbiology/s/data-availability) requires that you make available all data used to draw the conclusions arrived at in your manuscript. If you have not already done so, you must include any data used in your manuscript either in appropriate repositories, within the body of the manuscript, or as supporting information (N.B. this includes any numerical values that were used to generate graphs, histograms etc.). For an example see here: http://www.plosbiology.org/article/info%3Adoi%2F10.1371%2Fjournal.pbio.1001908#s5.

For manuscripts submitted on or after 1st July 2019, we require the original, uncropped and minimally adjusted images supporting all blot and gel results reported in an article's figures or Supporting Information files. We will require these files before a manuscript can be accepted so please prepare them now, if you have not already uploaded them. Please carefully read our guidelines for how to prepare and upload this data: https://journals.plos.org/plosbiology/s/figures#loc-blot-and-gel-reporting-requirements.

Upon resubmission, the editors will assess your revision and if the editors and Academic Editor feel that the revised manuscript remains appropriate for the journal, we will send the manuscript for re-review. We aim to consult the same Academic Editor and reviewers for revised manuscripts but may consult others if needed.

If you still intend to submit a revised version of your manuscript, please go to https://www.editorialmanager.com/pbiology/ and log in as an Author. Click the link labelled 'Submissions Needing Revision' where you will find your submission record. 

Sincerely,

Ines

--

Ines Alvarez-Garcia, PhD

Senior Editor

PLOS Biology

Carlyle House, Carlyle Road

Cambridge, CB4 3DN

+44 1223–442810

Reviewers' comments

Rev. 1

Ertekin et al report that fat body-specific knockdown of upd2, which was previously shown to be downregulated during starvation and increased with high-fat and high-sugar diets, increases feeding, decreases sleep, and enhances visual attention. Using a new tool for simultaneously assessing sleep and feeding-related behavior in individual flies in combination with a neuronal activity reporter, the authors show that activity of R4 neurons of the ellipsoid body are elevated in food-deprived and in upd2 mutants. Considering the prominent effect of sleep on cognitive function and the widespread practice of starving flies prior to behavioral tests, their finding that a manipulation that mimics starvation and suppresses sleep can improve performance on a cognitive task is novel, exciting, and may profoundly impact the way researchers design experiments and interpret their data. However, some of the conclusions in this manuscript are based on untested assumptions or only indirectly supported by the data. I believe that additional critical experiments and thoughtful rephrasing can greatly improve the cohesiveness and logical flow of this manuscript.

Major comments:

1) Some of the text used throughout the manuscript should be revised to more accurately reflect the findings. a) “Sleep need”: Data presented show that upd2 downregulation decreases sleep, but whether it also decreases sleep need per se has not been tested or shown. The authors should be conservative when referring to the reduced sleep phenotype of upd2 mutants and KDs. Alternatively, the authors could add additional experiments to support the idea, e.g., a thermogenetic approach could be used to show that flies do not sleep rebound after the downregulation has been lifted. b) Synonymously referring to upd2 as a “satiety signal” or downregulation of upd2 as “starved-like state”: Most of the manuscript does not directly compare the effect of upd2 manipulation with that of food deprivation (except in one instance, for R4 neuronal activity). I suggest avoid generalizing upd2 downregulation as “a lack of satiety cue” or “starvation-like state,” except in the Discussion.

2) Since the fat body and upd2 are central to the main findings of the manuscript, the driver and the RNAi should be validated. Fat body drivers are notoriously non-specific, and many publications use two or more fat body drivers to confirm a fat body-specific effect. Alternatively, or in addition, the authors could cite or show expression data for the driver. Regarding the RNAi, it’s common practice to use two independent RNAi constructs, and there are several other RNAi lines with no predicted off-target effects that are available for upd2, including another TRiP short hairpin. Finally, to more rigorously support the genetic interpretations, the authors could test whether fat body-specific expression of upd2 can rescue the feeding and sleep phenotypes in upd2 mutants.

3) Regarding the feeding estimates from the open-field platform, I think the authors are actually measuring a food-dwelling or food-proximity behavior, and the authors should be careful about calling it “feeding” or “feeding activity.” It certainly isn’t “food intake.” Other available methods of feeding estimation (e.g., FLIC, flyPAD) have undergone extensive efforts to validate that the estimates correlate with actual food consumption. Figure S3 is not evidence for the accuracy of feeding estimates, since it only shows that the measurements can distinguish behaviors in the presence and absence of food. Whether the assay can resolve different levels of food intake or feeding-related behaviors—and thus can be used for quantitative comparisons—needs to be tested. Some examples of additional experiments that may be useful include: a) Do the feeding estimates increase with prolonged starvation; b) Can the assay resolve a compensatory feeding response to food dilution; and c) Correlate their estimation with an actual feeding measurement (e.g., see the flyPAD paper, Figure 3 of Itskov et al. (2014) Nat. Comm.)

4) Figures 6 and S6: THIP’s sleep-inducing effect is validated in upd2 mutants but its effect on visual attention is tested in fat body upd2 knockdowns. The pharmacological validation and the phenotype measurements should be performed with the same genetic manipulation. Did upd2 KD flies also sleep more with THIP? Alternatively, did upd2 mutants also perform better than the controls in the visual attention task, with and without THIP? If the authors wish to support the idea that it is the starved state that enhances performance in the visual attention test, they need to show that food deprivation also improves visual attention in genetic controls. Without this validation, it is difficult to determine whether it is the “starvation-like state” or something entirely different that enhances visual attention of the upd2 KD flies.

5) R4 neurons as a neural correlate of starvation: This experiment doesn’t integrate well with the rest of the manuscript. The increased activity of R4 in the upd2 mutants is not connected to phenotypes other than starvation, including decreased sleep and enhanced visual attention. Also, because the relationship between starvation and increased R4 activity is not further studied, the experiment only raises more questions. I think excluding this section entirely would improve the cohesiveness of the manuscript. Alternatively, some suggestions for making this finding more substantiated and relevant include:

a) Rule out a contribution from sleep suppression on R4 activity. Both starved flies and upd2 mutants sleep less than controls. Although the authors show that the R4 activity is not significantly correlated with the range of sleep durations in freely-sleeping, well-rested flies, an “abnormal” level of sleep may still affect R4 activity. The authors should measure R4 activity in sleep-deprived controls and compare it to fed or starved controls and the upd2 mutants. For a more thorough dissociation experiment, the authors could additionally compare sleep-deprived/starved controls with sleep-deprived/fed or rested/starved controls to test whether the effects of sleep- and food-deprivations on R4 activity are additive.

b) To support the idea that R4 activity level correlates with nutrition state, the neuronal activity should be correlated with actual food intake instead of extended time spent on food, especially since the “feeding estimates” in the open-field arena has not been validated.

c) It is unclear why the feeding parameter being correlated with R4 CaLexA intensity is % time spent feeding rather than the number of feeding events, when the latter is the metric the authors use everywhere else when analyzing feeding behavior in the open-field arena (Figures 4, 7, S3, S8). Does the number of feeding events also correlate with R4 activity?

d) The relationship between feeding and R4 activity can be tested easily and more definitively using optogenetic stimulation of R4 neurons, similarly to Figure S5 in which the authors show that optogenetic activation of R4 decreases sleep. If R4 activity reflects starved state, does artificially activating R4 neurons increase food intake?

e) If the R4 neurons are THE neural correlates of starvation, and starvation is the cause of enhanced visual attention, then does R4 activation enhance visual attention in a similar way?

Minor comments:

1) Were virgin female flies used for all the experiments? It’s specified in some sections of the Methods but not in all.

2) Lines 132-133 (referring to Figure 1B), the authors write that upd2 mutants show “no significant change in total food consumption,” but from the data points shown it looks like the experiment was underpowered. The authors should not make any inference about the lack of “significant change” in this case.

3) Lines 138-139, the phrase “this suggests that upd2 mutants are feeding when they normally should be achieving most of their deeper sleep” implies that the flies are sleeping less during the period of hyperphagia because they are eating instead of sleeping. However, the sleep loss occurs mostly during the daytime instead of nighttime, and the latter is when the mutants overfeed. The statement should be rephrased or clarified.

4) Figure S1 does not “exclude the possibility of a circadian influence (Line 137).” The fact that the feeding pattern (nighttime hyperphagia) persists in constant dark actually suggests that the phenotype has a circadian component, per convention, as I understand it, in the field of chronobiology.

5) Does Figure 4D show a control fly, or a upd2 KD fly? Also, since heat plots are being used to help justify the feeding estimates, heat plots of both control and experimental flies should be shown for qualitative, visual comparison.

6) It has not been shown that upd2 acts through domeless in the Ilp2 neurons to modulate sleep, feeding, and visual attention phenotypes, as genetic interaction studies are missing. I advise the authors to either tone down the language or perform additional experiments to validate the genetic interaction. An ideal experiment would be to show that expressing upd2 in fat body rescues the behavioral phenotypes in upd2 mutants but not when dome is downregulated in Ilp2 neurons. Alternatively, the authors could show that dome RNAi has no additional effect on the behavioral phenotypes in the upd2 mutant background, although this study would not be as definitive.

Rev. 2

In this manuscript, Ertekin et al. uncover and investigate the link between Upd2 and sleep. The work rotates around the hypothesis that upd2 delivers a systemic satiety signal that, when missing, changes the feeding and sleeping behaviour of the flies.

The manuscript has several novel findings, with the main one being the characterisation of an unusual sleep phenotype of the Upd2 mutant and the altered feeding pattern, which shows a preference for nighttime feeding rather than during the day. Some of the findings are particularly intriguing: namely, the demonstration of increased activity in the R4 neurons in the brain of fed Upd2 mutant flies, which resembles that of a starved control fly, rather than that of a fed fly and the finding that upon knockdown of domeless (the Upd2 receptor) in Insulin-like peptide-2 neurons in the brain, the sleep and feeding phenotypes also resemble those shown with the Upd2 mutant.

However, I am afraid the main message of the paper is unfocused and the data are not convincing enough to support the main claim - if anything, they clearly reject it. Overall, the sensation is that there is strong disconnect between the line they authors chose to push and the actual results.

I find there are three main issues:

1) I am certainly not convinced that upd2 delivers a satiety signal. Simply put: a fly that is always hungry should eat more. upd2 mutant flies, instead, eat less and are smaller in size (however this latter is likely to be an not-discussed developmental effect).

Clearly there is a sleep phenotype during the day in udp2 mutants, and there is a different relationship with foraging but none of the results shown is actually compatible with lack of satiety signal - if anything, the opposite is true.

2) most of the putative "feeding analysis" in the paper rely on positional tracking. This is clearly the non-optimal tool for this story. Yes, flies seem to spend more time by the food but that does not mean they eat more (in fact, CAFE assay clearly shows they don't). Taken together, the results make me think that the udp2 phenotype is actually *not* a starvation phenotype: perhaps it is linked to food foraging for other reasons? flies use food to regulate their social interaction or to lay eggs. If the authors are interested in finding a mechanistically link, I suggest they look in that direction. If they want to continue on the starvation hypothesis I am afraid they would have to provide more convincing evidence. Perhaps using the flypad from the Ribeiro laboratory in Lisbon may shed some light.

3) the RNAi phenotype in the fat bodies does not recapitulate nor phenocopies the mutant phenotype, neither in terms of sleep nor in terms of foraging. This adds a level of confusion that leaves the reader disoriented, especially because the authors claim otherwise.

4) I don't know what to make of the increase in attention phenotype shown in figure 6. Perhaps is interesting but, again, it is based on too many assumptions. One important assumptions the authors make is that because upd2 flies sleep less than control, they may be sleep deprived (line 421). I would argue the opposite is true: they sleep less because they need less sleep. When we are comparing two different genotypes, the fact that one sleep less than the other says nothing about sleep deprivation state. So in conclusion, ok that udp2 KD have a slightly better attentive performance but how does this relate to sleep at all?

Additional comments on figures are as follows.

Figure 1

• upd2 mutant flies are smaller (how smaller? Please do not make quantitative claims without quantifying). A comparison should be made between the Upd2 mutant line with control flies which have been matched for size to rule out any effect that may result from a smaller and slimmer fly just showing a different feeding pattern or sleep pattern.

• An additional control of wild-type flies which have been starved could serve as point of reference for the claims.

Figure 2

• The controls used, particularly those in Figure 2A, B and C seem to show significant differences between each other - please include full statistics.

Figure 3

• I am personally uneasy about dissecting sleep parameters such as sleep bout lengths because we have no idea whatsoever of what they mean - if anything. I understand the field for some reasons like those, so it's fine to have these measures. However, I do not think one can talk about "sleep quality" as a substitute for longer bouts. This is a tautology that we should not promote in the field. The authors do this multiple times in the manuscript. In fact, their argument is that udp2 flies have worst sleep quality but at the same time they show they perform better in cognitive tests.

• In figure 1K-L, the authors show that udp2 mutant flies have shorter bouts. If, like the mutant line, the knockdown line shows bout lengths similar to that of the mutant, measuring arousal past 31 minutes during the day would lead to measurements being taken from a non representative sample of bouts, as no individuals from the mutant line had bout lengths longer than 20 minutes (similarly with measurements in the night time). Incidentally, the sample size used for these experiments is also not stated in the manuscript.

This figure would benefit firstly from the same analysis done with the mutant line where measures of bout duration and number of sleep bouts are taken first before measuring arousal.

Figure 4

• The assay used here for feeding is not state-of-the-art and is confounded by too many issues. See major point #2

Figure 5

• This is interesting. Figure 5D would benefit from adding a starved version of the Upd2 mutant in addition to the fed version. Also, quantitative palettes should be used so that the highest level is never reached and GFP intensity appears to be almost saturated in this image - it would be interesting to see if the starved Upd2 mutant could give an even greater intensity.

Supplementary Figure 6

• This figure would benefit from quantification of food intake with the drug. If the drug also changes feeding behaviour or increases how much food the flies eat, they may show greater sleep from a postprandial response.

• If using Gaboxadol changes the amount of sleep but does not change attention, this could be further support for the phenotype being one of increased foraging rather than a starvation phenotype.

Rev. 3: Alex Keene - please note this reviewer has waived anonymity

This is a very creative and innovative manuscript that describes a relationship between sleep and attention. A number of conceptual aspects are highly novel and will broadly influence the field including the identification of upd2 as a molecular link between peripheral energy stores and central brain regulation of sleep, and the finding that increased arousal in upd2 mutants may promote attention, despite a reduction in total sleep. The manuscript is well written and technically sound. I do believe there are a number of addition experiments that are needed to support or clarify the conclusions in particular see points 2,3, 4, 6, 7.

1. While there is evidence that upd2 functions as a homolog of leptin (Rajan et al, 2012, Rajan et al, 2017), my understanding is this is still relatively controversial and it may be safer to describe upd2 as a secreted cytokine, then discuss its leptin-like role in the discussion section.

2. The feeding data in 1B are non-significant, but the absolute values are different by nearly 50%, suggesting the need for increased statistical power. Also, the legend states N=40, but there are only 5 data points. Please clarify.

3. Line 183. ‘any effects on sleep were inconsistent compared to controls.’ It is unclear what this means. The best control is the RNAi line alone, which provides a phenotype, raising the possibility that upd2 functions in the brain. It may be worth testing with additional GAL4 or fat body drivers to clarify these results.

4. The authors bin arousal threshold in 1-30 or 30-60 minute bins, but I believe their earlier work has shown that the switch from light to dark sleep occurs early. The conclusion that sleep intensity is reduced in upd2 mutant flies would be improved by greater temporal resolution.

5. Based on the text, I do not understand the contributions of Figure 4. First, other systems have been devised to simultaneously measure sleep and feeding e.g. ARC assay. Second, the system cannot distinguish whether the animal is eating, only whether it is near the food which can be detected using the DART. If the purpose is to confirm results in a different arena, then this is valuable and should be stated. Otherwise, please clarify how this improves previous figures.

6. In figure 5 it would be useful to show CaLexA in the fed and starved state for both the mutant and wild-type. In addition, this would be useful for the R4 correlations in 5B.

7. Why was the attention assay only performed in RNAi knockdown flies. It is incongruous to have the R4 experiment exclusively in mutants and the attention work in knockdown experiments. The easiest way to rectify would be to perform the attention experiments in mutants. I also do not understand S6. The sleep measurements with THIP are in mutants, but the attention assay is in the RNAi knockdown.

Minor comments

1. Line 54: ‘the phrase ‘maintain well-tune cognitive processes’ is vague.

2. The first paragraph lacks citations. Even though the concepts are generally well-accepted it would be helpful to point readers to relevant literature.

3. It’s not clear why the day and night are labeled by yellow and blue. Especially since others have used this to denote fed-starved. May be better to use white and black boxes to maintain consistency with the circadian literature.

4. Please clarify line 183. If they are awake/foraging, why not feed?

5. In methods, please reference the w+ line used for outcrossing.

Rev. 4

In this manuscript, the authors present an interesting set of experiments that seek to understand the relationships among sleep, physiology, and cognitive function. While we find the questions highly relevant and the data intriguing, there are several conceptual concerns that limit our enthusiasm for the manuscript in its present form. Some involve what we believe to be over interpretation of the data, which, in our opinion, obfuscate the main thesis of the paper and logical thread of the work as well as confuses direct observation with interpretation, thereby potentially limiting its overall impact. Other concerns center on what might be considered insufficient depth of mechanistic understanding of key components of their model. Significant textual revisions are recommended, as are a small number of additional experiments.

The title of this manuscript implies discovery of a link between the expression of a satiety signal (i.e., upd2) and sleep need and visual attention. These reviewers agree that this is a well-supported and interesting finding. However, in our opinion, this message is diluted throughout the manuscript by the authors persistent presentation of conjecture as established fact (e.g., by equating upd loss with starvation), by conflating key ideas (e.g., sleep need is not the same as sleep loss), and by over-interpretation of behavioral data (e.g., positional preference does not imply feeding). These positions are unnecessary in the Introduction and Results, and they actively hinder the reader’s ability to carefully interpret the experiments. Such conjecture of the data should be confined to the Discussion.

Assertion of a starvation-like state: The authors repeatedly state that loss of upd signaling emanating from the fat body is responsible for a “starved-like” state; a state the authors believe mimics the starvation-induced sleep deprivation phenotype widely reported by others. Maybe, maybe not. True, some phenotypes of upd loss are similar to those in starved animals (e.g., decreased sleep duration and quality). But others are not (e.g., upd LOF flies don’t eat more, starved flies do). The timing of when upd mutants feed is altered, but this may indicate changes in sleep-wake behavior and/or in baseline metabolic state. We agree that it is intriguing to consider that these animals are “genetically starved,” but we are not convinced this is the case. We propose two possible ways to address this. First, the authors could simply move the discussion of a possible starved-like state and its implications to the Discussion and maintain a focus on describing the results precisely (e.g., that upd loss is causing the phenotypes, not starvation). Whether the flies are starved does not impact the fact the sleep and attention are affected by upd. Alternatively, the authors might provide more convincing evidence of perceived starvation such as:

(1) Determining whether starving upd mutants or yolk-GAL4 > upd-RNAi flies alters sleep, feeding, and attention phenotypes: if loss of upd achieves a starved-like state, then further starvation would not significantly alter these phenotypes

(2) Asking whether conditional loss of upd function using tub-GAL80ts or tub-GeneSwitch affects baseline sleep, feeding, and attention measurements relative to baseline prior to the manipulation. Furthermore, this strategy would also help clarify whether the phenotypes reported in the paper are caused by the current state upd signaling or instead cause by loss of upd signaling throughout development.

Position preference as an “accurate estimate” of feeding: Similar to the claim of genetic-starvation, the authors use positional preference to infer “feeding events.” These two may indeed be correlated, although is many instances they are not. Either way, the authors should simply state these data as “positional preference” or “time spent near the food” in the results to avoid the indication that this assay explicitly measures feeding. The lack of flies coming to the food cup when it is covered with parafilm does not provide evidence for or against what they do when they are there, and the observation that they spend similar amounts of time there when it is filled with agar seems to us to argue against the notion that this is an accurate measure of feeding. Either way, its accuracy has not been established.

Lack of connection between DILP2 and R4 Neuron Activity: Given the observations presented in Figures 5, 7, and associated supplemental figures, DILP2 neurons must somehow communicate with R4 neurons to regulate their excitability and, in turn, form the basis of nutritional state control over sleep behavior. The manuscript would be substantially strengthened if the authors provided data illuminating the relationship between these neurons, which can be done either by (1) neuroanatomical characterization of possible synaptic connections made between DILP2 and R4 neurons or (2) determining whether R4 neurons respond to artificial activation/inhibition of DILP2 neuron activity using the CaLexA approach.

Phenotype in mated female and/or male flies: It is worth noting that only virgin female flies were tested throughout the manuscript. However, it is widely understood that sleep/feeding behavior is not only sexually dimorphic, but is also strongly modulated by the reproductive status of female flies. The authors should strongly consider investigating the upd mutant phenotypes in mated females and male flies to better establish the relevance and generality of the observations.

Requirement of amino acid for feeding/sleep phenotype: This reviewer noticed that most of the feeding measurement were performed using 5% sucrose food. However, sucrose-only food does not support normal lifespan or physiology and there is increasing evidence supporting the importance of specific amino acids influencing feeding-dependent effects on sleep-wake regulation and homeostasis. While it is, perhaps, extreme to ask that these experiments be repeated under standard nutritional conditions, this limitation should be clearly stated and discussed in the Discussion section.

Differences between mutant and tissue-specific RNAi knockdown observation: The authors generally conclude that upd signaling from the fat body is responsible for its effects on feeding and sleep (lines 200-201; Fig 2). However, liquid feeding amount (e.g., daytime feeding) and sleep measurements (e.g., modest effects during nighttime; not significant across all controls) appear to be different between the mutant and fat-body knockdown strategies (Fig 1C vs. Fig 2A). The authors should comment on this discrepancy. As a side note, data showing that the upd2-RNAi is effective should be presented, either by determining whether ubiquitous knockdown of upd2 phenocopies the mutant phenotypes and/or by measuring the extent of upd transcript reduction caused by fat-body specific RNAi knockdown.

Sleep fragmentation measurements using tissue-specific RNAi approach: The authors should quantify and present sleep fragmentation parameters, similar to what is presented for the upd mutant in Fig 1H-L, for the tissue-specific knockdown data collected in Fig 2, 4A-C, and 7C-D. The expectation is that night-time sleep is much more fragmented when upd is knocked-down in fat body cells (i.e., increased sleep bout number and decreased average bout duration).

Additional Comments:

The Fig 1 legend states that “(B) Food intake was measured with Café assay, with 5 flies/chamber over 20 hours (n=40 per genotype)”. However, the number of data points do not match this description (only 5 data points appear to be plotted). The only possible explanation is if each data point represents a unique experiment. If that is indeed the case, that should be explicitly stated.

In the food intake figure presented in Figure S1, the term “ZT” along the x-axis should be changed to “CT” since the experiment is being performed under constant darkness conditions.

Both the interpretation and presentation of Figure 7 should appear before Figure 5 for the sake of continuity. According to the authors’ proposed model, upd signaling from the fat body is received by DILP2 neurons, which in turn somehow sends direct/indirect nutrient status information to R4 ring neurons to regulate sleep. The way it is currently presented in the manuscript, it seems like the R4 ring neurons come before the DILP2 neurons.

Additional detail should be provided in the Methods section regarding measures of sleep intensity, specifically the calculation/algorithm that was used to determine whether a fly responded to a stimulus.

---

## [Decision Letter · Decision Letter 2]

11 May 2020

Dear Bruno,

Thank you very much for submitting a revised version of your manuscript "Downregulation of a cytokine secreted from peripheral fat bodies improves visual attention while reducing sleep in Drosophila" for consideration as a Research Article at PLOS Biology. Please accept my apologies again for the delay in sending you our decision. This revised version of your manuscript has been evaluated by the PLOS Biology editors, the Academic Editor and three of the original reviewers.

As you will see, two of the reviewers feel that the manuscript is greatly improved and only have a few minor suggestions to clarify some points. Reviewer 2, however remains unconvinced about the feeding/starvation aspect of the manuscript. After discussing these comments with the academic editor, we do feel that while it is fine to propose in the discussion that the upd2 mutants represent a starve state, you should refocus and temper your message, acknowledging all the caveats associated with this hypothesis and not taking it as a given making that the starting point. In addition, you need to address Reviewer 2’s Point 3, which we consider important.

In light of the reviews (attached below), we are pleased to offer you the opportunity to address the remaining points from the reviewers in a revised version that we anticipate should not take you very long. We will then assess your revised manuscript and your response to the reviewers' comments and we may consult the reviewers again.

We expect to receive your revised manuscript within 1 month.

**IMPORTANT - SUBMITTING YOUR REVISION**

*Resubmission Checklist*

*Published Peer Review*

*PLOS Data Policy*

*Blot and Gel Data Policy*

Sincerely,

Ines

--

Ines Alvarez-Garcia, PhD

Senior Editor

PLOS Biology

Carlyle House, Carlyle Road

Cambridge, CB4 3DN

+44 1223–442810

Reviewers’ comments

Rev. 1:

The revised manuscript is greatly improved. Removing the opto experiments and the changes to the text make the paper much more readable. The authors have addressed the critical concerns adequately. I have only two minor suggestions:

1) Page 17, line 459: I don't think this comparison should be made across experiments. If you really want to make this claim, I think the lines need to be tested at the same time, both with and without THIP.

2) Page 18, line 484-5: The subheading overstates the findings. I think the subheading should be revised to be more conservative, similar to the figure legend title.

Rev. 2:

Main issues:

Point 1) I am certainly not convinced that upd2 delivers a satiety signal. Simply put: a fly that is always hungry should eat more. upd2 mutant flies, instead, eat less and are smaller in size (however this latter is likely to be an not-discussed developmental effect). Clearly there is a sleep phenotype during the day in udp2 mutants, and there is a different relationship with foraging but none of the results shown is actually compatible with lack of satiety signal - if anything, the opposite is true.

AU answer: Increased activity of the R4 neurons is compatible with a starvation signal. This aligns with previously published work (Park et al., 2016). However, we concede that before this result, the Reviewer is correct in questioning our terminology, so we have changed our language in the text accordingly and reserved the question of satiety signals for the discussion. The Reviewer suggests that upd2 mutants should be eating more. First, flies probably eat as much as they can, so there may be a ceiling effect. What is more interesting here is if they display more feeding-related behaviours. This is exactly what we find, when we move beyond the Café assay and look at feeding-related behaviour in knockdown animals in our open field arena. In upd2 and domeless knockdown animals, the number of feeding events 10 increases during the day and the night (see Fig. 4F and Fig. 7E). It is however only significant against both genetic controls for nighttime feeding. The data in Fig. 4F quite convincingly show a generalized increase in feeding-related behaviour, even if one daytime control is not significant. We do not know why the Café assay did not show a similar increase, but again this may be due to a ceiling effect, or may have to do with the food quality (sugar water versus standard fly food), or the finer-grained behavioural analysis available for individual flies in our open field feeding assay (see discussion below regarding this assay). The important conclusion is that the manipulated flies are displaying more feeding behaviour. There is only one experiment where this does not hold, and that is daytime Café for the upd2 mutants (Figure 1B), but in total darkness food consumption in the mutant trends to more (Fig. S1). Everywhere else, upd2 manipulations shows equivalent or more feeding behaviour

Reviewer’s answer: True you show that one circuit namely R2 neurons which respond to starvation have increased activity. There are many many more circuits involved. I refer you to Krishna Melnattur, Paul Shaw 2019 review on the subject. What about LHLK neurons? PAMs? IPCs/DILPs, NPF? I would expect a much more comprehensive testing of your hypothesis. As it stands it is a big (yet in my opinion unnecessary) part of the paper which is not adequately addressed. (see general comments at the end for clarification)

Point 2) most of the putative "feeding analysis" in the paper rely on positional tracking. This is clearly the non-optimal tool for this story. Yes, flies seem to spend more time by the food but that does not mean they eat more (in fact, CAFE assay clearly shows they don't). Taken together, the results make me think that the udp2 phenotype is actually *not* a starvation phenotype: perhaps it is linked to food foraging for other reasons? flies use food to regulate their social interaction or to lay eggs. If the authors are interested in finding a mechanistically link, I suggest they look in that direction. If they want to continue on the starvation hypothesis I am afraid they would have to provide more convincing evidence. Perhaps using the flypad from the Ribeiro laboratory in Lisbon may shed some light.

AU answer: The Reviewer is correct in suspecting that feeding-related behaviours are affected. This is exactly what our new assay is meant to detect: feeding events are not only about dwelling near the food cup, there are a number of other movement criteria that have to be satisfied to count as a feeding event (detailed in the Methods). We have now provided starvation data to further validate our assay, showing a doubling of the number of feeding events in starved flies (Fig. S7B). While it is true that we do not know how much food is actually ingested, we would counter that does not matter because it is the feeding behaviour (visits to the food cup) that is of interest to this study and aligned with our altered attention phenotypes. Indeed, that is the direction our study takes (visual attention behaviour), rather than the metabolism angle. We nevertheless feel confident that our feeding-related metric correctly estimates food consumption. The Reviewer might want to compare the behavioural estimates in Figure 4F with food consumption in Figure 2A (for upd2 knockdowns), or Figure 7E with Figure S11 (for domeless knockdowns). The graphs seem convincingly similar to us.

Reviewer’s answer: I have a problem with the feeding aspect of the paper in general. Please see other comments that address this concern

Point 3) the RNAi phenotype in the fat bodies does not recapitulate nor phenocopies the mutant phenotype, neither in terms of sleep nor in terms of foraging. This adds a level of confusion that leaves the reader disoriented, especially because the authors claim otherwise.

AU answer: We are puzzled where the Reviewer is confused. Figure 2C (knockdown sleep data) does phenocopy the mutant sleep data (Figure 1G): day-time sleep is severely reduced and night time sleep only lightly. That effect sizes are not as large in the knockdown is probably attributed to penetrance. But the effect is the same. By foraging, the Reviewer is probably referring to the Café results? Here it is true that there is some discrepancy, with less daytime feeding. But total food consumption is similar to controls, also in DD trials (Fig. S1). There is no discrepancy at all between the upd2 and domeless knockdown results, for any phenotype.

Reviewer’s answer: There is a problem with replicating feeding results in this paper. In figure 1a Upd2 mutants have nighttime hyperphagia but no overall effect over 24hrs. In Figure 2a there is an overall difference between one control and KD but not the other in total intake. In figure 4E KD is different from one control group but not the other and in 4F KD are hyperphagic at both day and night. I am struggling to draw a conclusion here. Also from what I can tell/ it is not clear if you are using different diets in CAFÉ assay/open field arena and in DART assay. Are these comparable- please clarify?

Point 4) I don't know what to make of the increase in attention phenotype shown in figure 6. Perhaps is interesting but, again, it is based on too many assumptions. One important assumptions the authors make is that because upd2 flies sleep less than control, they may be sleep deprived (line 421). I would argue the opposite is true: they sleep less because they need less sleep. When we are comparing two different genotypes, the fact that one sleep less than the other says nothing about sleep deprivation state. So in conclusion, ok that udp2 KD have a slightly better attentive performance but how does this relate to sleep at all?

AU answer: The Reviewer is correct to doubt that the upd2 mutants are sleep deprived. This is also what we thought, which is exactly why we tested this assumption by providing them with more sleep (via the sleep drug THIP), to see if this ‘corrected’ their hyper focussed attention phenotype. It did not. This is an important result (now Fig. 7). If their attention phenotype was a maladjustment due to insufficient sleep, then providing mutants with more sleep using this method might have corrected it, as shown previously (Kirszenblat et al, 2018). Rather, upd2 mutants seem to need less sleep, and one consequence is to be hyper focussed. This is in marked contrast to sleep-deprived wild-type flies, which are less focussed (more distractible), which we reported in Kirszenblat et al, 2018. That upd2 mutants do the opposite while also sleeping less is precisely what makes our study interesting. We have tried to clarify this striking result a bit better in the discussion (lines 577-599).

Reviewer’s answer: Ok- I understand that people like to use this drug but I doubt it recapitulates natural sleep. It may be more akin to anaesthesia. I don’t think this experiment adequately addresses the issue of sleep deprivation but accept this is a tool used in the field.

Additional comments on figures:

Figure 1

• upd2 mutant flies are smaller (how smaller? Please do not make quantitative claims without quantifying). A comparison should be made between the Upd2 mutant line with control flies which have been matched for size to rule out any effect that may result from a smaller and slimmer fly just showing a different feeding pattern or sleep pattern.

• An additional control of wild-type flies which have been starved could serve as point of reference for the claims.

AU answer: Physical difference in the mutants have been described previously (Rajan & Perrimon, 2012) and we see the same. We did not feel the need to replicate these measurements, as we trust these are the same mutants.

Reviewer’s answer: Ok.

Figure 2

• The controls used, particularly those in Figure 2A, B and C seem to show significant differences between each other - please include full statistics.

AU answer: We have indicated the significant effects, if there were any. The full statistics can be found in our deposited files. The specific statistics refered to are in the metadata provided for Figure 2.

Reviewer’s answer: Ok

Figure 3:

• I am personally uneasy about dissecting sleep parameters such as sleep bout lengths because we have no idea whatsoever of what they mean - if anything. I understand the field for some reasons like those, so it's fine to have these measures. However, I do not think one can talk about "sleep quality" as a substitute for longer bouts. This is a tautology that we should not promote in the field. The authors do this multiple times in the manuscript. In fact, their argument is that udp2 flies have worst sleep quality but at the same time they show they perform better in cognitive tests

AU response: The Reviewer may not have understood our sleep intensity measures in Figure 3, which follow from multiple publications (that we reference) where these are more thoroughly explained (e.g., Faville et al, 2015). Briefly, these are not simply sleep duration bouts. We measure sleep intensity by probing for behavioural responsiveness to mechanical stimuli, and this can happen any time between 0 and 60min. How responsive they are (how deeply asleep they are) depends to some extent on how long they have been asleep, especially at night (see Fig. 3E). The binned data in Fig. 3B is just to show that flies were probed for all of the different sleep durations. There are comparatively fewer probing events for 30-60min during the day because fewer flies ever sleep that long during the day, on average. Our measures stop at 60min because our probing events are hourly; how long flies have been asleep before the hourly probe is stochastic. We now provide some additional Methods (lines 667-672). We hope we have clarified better our sleep intensity assay, so that the reader can appreciate why allude to sleep quality. There is no question that upd2 mutants and knockdown animals have more fragmented sleep (Fig. 1J-L) and are easier to wake up (Fig. 3D,E). Following Reviewer #4’s questions (below), we now provide the sleep consolidation / fragmentation data for the upd2 and domeless knockdown strains as well (Figures S5 and S10). It all quite consistently shows a difference in sleep quality in these animals. That this should be matched by ‘improved’ attention is again precisely what makes this study interesting. We are however cautious in not interpreting this as ‘better’ performance, and provide quite a bit of discussion on what this ‘improved’ attention phenotype might mean ethologically (lines 577-599).

Reviewer’s answer: Line 243: Authors say that flies can compensate for sleep loss by exhibiting deeper sleep during shorter sleep bouts yet use a vibration stimulus to probe “sleep depth”. Sleep depth does not necessarily equate to increased quality. For example in mammals sleep is characterised by different stages (REM, non REM 1,2,3,4) which are necessary for different aspects of cognitive performance. REM sleep is though to be important for dissociation between memory and emotional salience for instance, but it is much easier to wake someone in REM sleep compared to non rem stages. Please provide some discussion/caveats to your conclusions here.

• In figure 1K-L, the authors show that udp2 mutant flies have shorter bouts. If, like the mutant line, the knockdown line shows bout lengths similar to that of the mutant, measuring arousal past 31 minutes during the day would lead to measurements being taken from a non representative sample of bouts, as no individuals from the mutant line had bout lengths longer than 20 minutes (similarly with measurements in the night time). Incidentally, the sample size used for these experiments is also not stated in the manuscript. This figure would benefit firstly from the same analysis done with the mutant line where measures of bout duration and number of sleep bouts are taken first before measuring arousal.

AU answer: Sleep in upd2 mutants was more severely affected compared to the knockdown strain. This is not surprising as the knockdown experiments were performed to circumvent developmental effects (Yolk-Gal4 expression is only in adults). As can be seen in Fig. 3B, knockdown flies are sleeping past 31 minutes, even during the day. There are fewer instances though, as expected, but enough to say something about sleep intensity differences for most time bins. The dataset in Fig. 3 are from the same flies as in Fig. 2, which we have noted in figure legend. We have now added the sample sizes under this figure as well.

Reviewer’s answer: ok

General comments from the reviewer:

In general there are some interesting aspects to this paper but the feeding/starvation aspect of it is confusing and weak. For me a simpler and stronger story could be compiled by looking at the effect of upd2 and domeless mutants/KD on sleep, depth and fragmentation. The apparent lack of compensation for sleep loss yet maintenance/improvement in attention in domeless and upd2 mutants/KD animals is the most interesting part. You can discuss feeding dysregulation/starvation in discussion but I don’t think you have adequately demonstrated that you are mimicking a starved state or feeding dysregulation in these animals. As I said previously you are not using state of the art means to measure intake (such as ARC or flypad). In general the feeding phenotypes are weak, upd2 mutants are different sizes and as far as I can tell you do not normalise food intake by body size.

I would encourage a rework of manuscript composed of figure 1F-L, Figure 2 C,D,G,H,K,L. Figure 3, Figure 5 (including comparison between control and Upd2 fed conditions but not starved.), Figure 6 and Figure 8. It still works well without feeding side of it.

Other comments

Images in Figure 5D are oversaturated. While the effect is clear from the representative images it makes me nervous about the validity of the imaging technique as a whole.

Where are stats/Ns in 3B? Are all bouts observed in longer bins contributed by one fly? Or many? Ns/bin not reported on graph/ legend, only the number of flies in the experiment in figure legend. “(B) The number of animals in each sleep duration bin for both day and night were similar in both genetic controls (black and grey) and in knock-down flies (orange).” How can this be if in figure 1 you show strong differences in bout number and length? Please clarify. Incidentally if the genotypes are not receiving the same number of stimuli (stimulus/fly/genotype), it is unlikely that they are receiving the “same” stimulus as you may have a problem with habituation. You will need to acknowledge this if this is the case.

Line 325: why was 30s chosen as criteria for feeding? What is average feeding bout length? Are you not missing shorter feeding bouts? This is why a method like flypad would be far superior in this study.

Rev. 3: Alex Keene

The revised version of this manuscript has addressed all my initial concerns through the inclusion of additional data and changes within text. I have two minor suggestions below that could be addressed at the authors discretion.

1. I prefer the sentence on line 83 to be deleted or modified. Behavioral performance is used in a non-specific way, and one could envision many behaviors (such as foraging) that are better when the animal is not well-fed.

2. Clarity could be added to paragraph on insulin signaling (line 616). Which neurons were found to be Ilp2 targets in larvae? Does the suggestion they communicate 'indirectly' mean non-synaptically or through intermediary neurons? It is also worth noting that there are many other sources of insulin, other than Ilp2 neurons.

---

## [Editor Report · Decision Letter 3]

9 Jun 2020

Dear Dr van Swinderen,

Thank you for submitting your revised Research Article entitled "Downregulation of a cytokine secreted from peripheral fat bodies improves visual attention while reducing sleep in Drosophila" for publication in PLOS Biology. I have now checked the revision and obtained advice from the Academic Editor.

We're delighted to let you know that we're now editorially satisfied with your manuscript. However before we can formally accept your paper and consider it "in press", we also need to ensure that your article conforms to our guidelines. A member of our team will be in touch shortly with a set of requests. As we can't proceed until these requirements are met, your swift response will help prevent delays to publication. Please also make sure to address the data and other policy-related requests noted at the end of this email.

*Copyediting*

*Published Peer Review History*

*Early Version*

*Submitting Your Revision*

Sincerely,

Ines

--

Ines Alvarez-Garcia, PhD

Senior Editor

PLOS Biology

Carlyle House, Carlyle Road

Cambridge, CB4 3DN

+44 1223–442810

DATA POLICY:

Many thanks for providing a data file containing all the data underlying the graphs displayed in the figures. I have checked the file and I would like you to please fulfil the following requests regarding the data you provided:

- Label clearly the data in Fig. 2 that belongs to each panel and add the data for Fig. 2J, K, L

- Add missing data from Fig. 4B

- Relabel the data in Fig. 8 (currently labelled as Fig. 7) and add missing data from Fig. 8C

- Add all the values of the graphs in Fig. S2A, B – currently only the mean is shown

- Add missing data for Fig. S7D, E, F, G – there is one panel wrongly labelled as S7C, so please add the right label and the data for the missing sections.

In addition, please also ensure that the figure legends in your manuscript include information on WHERE THE UNDERLYING DATA CAN BE FOUND, and ensure your supplemental data file/s has a legend.

---

## [Editor Report · Decision Letter 4]

13 Jul 2020

Dear Dr van Swinderen,

On behalf of my colleagues and the Academic Editor, Amita Sehgal, I am pleased to inform you that we will be delighted to publish your Research Article in PLOS Biology. 

Early Version

PRESS 

Kind regards,

Pamela Berkman

Publishing Editor

PLOS Biology

on behalf of

Ines Alvarez-Garcia,

Senior Editor

PLOS Biology